# Microglia contribute to neuronal synchrony despite endogenous ATP-related phenotypic transformation in acute mouse brain slices

Péter Berki[1,2,3,12], Csaba Cserép [4,12], Zsuzsanna Környei [4,12], Balázs Pósfai [4], Eszter Szabadits[4], Andor Domonkos [4,5], Anna Kellermayer[4], Miklós Nyerges [4], Xiaofei Wei [6], Istvan Mody [6], Araki Kunihiko[7,8], Heinz Beck[7,8], He Kaikai[9], Wang Ya[10], Nikolett Lénárt[4], Zhaofa Wu [11], Miao Jing[10], Yulong Li [9], Attila I. Gulyás [2] & Ádám Dénes[4]✉

Acute brain slices represent a workhorse model for studying the central nervous system (CNS) from nanoscale events to complex circuits. While slice preparation inherently involves tissue damage, it is unclear how microglia, the main immune cells and damage sensors of the CNS react to this injury and shape neuronal activity ex vivo. To this end, we investigated microglial phenotypes and contribution to network organization and functioning in acute brain slices. We reveal time-dependent microglial phenotype changes influenced by complex extracellular ATP dynamics through P2Y12R and CX3CR1 signalling, which is sustained for hours in ex vivo mouse brain slices. Downregulation of P2Y12R and changes of microglia-neuron interactions occur in line with alterations in the number of excitatory and inhibitory synapses over time. Importantly, functional microglia modulate synapse sprouting, while microglial dysfunction results in markedly impaired ripple activity both ex vivo and in vivo. Collectively, our data suggest that microglia are modulators of complex neuronal networks with important roles to maintain neuronal network integrity and activity. We suggest that slice preparation can be used to model time-dependent changes of microglia-neuron interactions to reveal how microglia shape neuronal circuits in physiological and pathological conditions.

Since its first applications several decades ago[1,2], the acute brain slice preparation technique has become a key tool in the field of neuroscience and has extensively contributed to our understanding of cellular physiology. As the methodology of slice preparation went through numerous iterations over the years[3–5], improvements have led to acute slices even capable of producing spontaneous oscillations, similar to those observed in vivo[6–9]. Meanwhile, ex vivo studies of non-neuronal cell types such as astrocytes and microglia have also been emerging[10,11], alongside with increasing focus on bidirectional communication between neurons and glial cells[12,13]. For instance, the

release of gliotransmitters, such as ATP, adenosine or glutamate, by astrocytes has been shown to influence neuronal activity[14–16]. Microglia are the primary immune cells of the CNS parenchyma, with essential roles beyond their immune function under both physiological and pathological conditions[17–19]. Because microglia are heavily involved in brain development and maintenance of neuronal populations[20,21], while their functional alterations are linked to a wide range of human diseases[22–24], interest in understanding microglial function has substantially increased over the last decade. As such, acute slice preparations also proved to be an essential tool for the investigation of

microglia, with better preservation of microglial phenotype resembling their physiological states than the in vitro culture system[25–29], based on robust transcriptomic, proteomic and functional evidence[10,27,30–36]. However, it is also acknowledged that microglia can become reactive in slices and transform into an amoeboid phenotype[37–39]. Importantly, while the acute slice technique has contributed to understanding microglial physiology[10,30,32,35,40], the impact of methodological practices, relevant time frames, mechanisms influencing microglial states upon slice preparation, and most importantly, the impact of microglial function on electrophysiological measurements of neurons have remained vaguely characterized. Despite the known sensitivity of microglia to even subtle changes in their microenvironment[18,41] mechanistic data from the first few minutes to hours after slice preparation (a relevant time frame for most studies) with appropriate temporal resolution to assess microglial states are not available at present. Thus, it is currently unclear how time-dependent changes in microglia may influence complex neuronal circuits and their reorganization in acute slice preparations. To this end, we set out to investigate microglial phenotype changes and their impact on neuronal networks simultaneously, in slice preparations capable of producing spontaneous network activity by using an experimentally relevant timeframe. For these studies, we made use of established protocols and the extensive expertise in slice preparation available at three independent research laboratories, which pioneered the preparation of high-quality brain slices and introduced technical developments enabling the first investigation of complex network oscillations[6,42–44]. Our results suggest that microglia maintain fundamental features of neuronal networks in acute slice preparations in a P2Y12R and CX3CR1-dependent manner, which is influenced by rapid and robust microglial phenotype changes due to injury-related acute ATP release and sustained focal ATP dynamics. These observations may be of importance for all acute brain slice studies investigating either neuronal function or microglial responses.

## Results

### Microglia gradually migrate towards the surface of acute slice preparations in a P2Y12R- and CX3CR1-dependent manner

Microglia are well known to react rapidly to injury or tissue disturbance in the brain parenchyma[45,46]. To investigate how injury caused by acute slice preparation influences microglial functions at the population level, we first tracked microglial cell body and process distribution during a 5-h incubation period in 300 μm-thick acute hippocampal slices from CX3CR1[+/GFP] microglia reporter mice (P35 days). We used a strictly controlled preparation and incubation procedure across all measurements (see details in the "Methods" section), optimized for studying spontaneously occurring sharp wave-ripple activity (SWR)[6,47]. To monitor time-dependent changes, slices were immersion-fixed at different timepoints after cutting (Fig. 1a). Subsequently, preparations were re-sectioned and mounted onto glass plates for analysis (Fig. 1b). Only the native microglial GFP signal was imaged via confocal laser-scanning microscopy (Fig. 1c). We found that translocation of microglial cell bodies occurs rapidly after slice preparation with most extensive changes in the top region (~40 μm) of slices (Fig. 1d). Here, the density of cell bodies gradually increased by 75% throughout the 5 h of the incubation (two-way ANOVA, $p < 0.001$). Top region of slices showed a 42% higher density of cell bodies when compared to the bottom region ($p < 0.01$) after 2 h of incubation, a relevant timepoint for most electrophysiological measurements (Fig. 1e, black statistical indication). The total number of microglia located in whole cross sections did not differ significantly during the experiment (43.07 cells/grid at 0 min, 41.50 cells/grid at 5 h; $p = 0.788$), suggesting that microglia loss does not contribute to the observed changes in cell distribution. To test whether two prominent microglial signalling pathways could play a role in cutting-induced microglia displacement, we repeated the experiments in the presence of a potent and selective

P2Y12-receptor (P2Y12R) inhibitor, PSB0739 (PSB) or using slices from fractalkine-receptor (CX3CR1) knockout mice[48]. We found that PSB did not change substantially the cell body dislocation of microglia, while dislocation towards the surfaces was markedly impaired in slices from CX3CR1-KO mice ($p < 0.0001$; Fig. 1f).

Microglial processes can move two orders of magnitude faster than microglial cell bodies[49], thus we went on to analyse the possible rearrangement of these processes. The spatial distribution of microglial processes within the slice showed similar, but even more pronounced alterations during the incubation process (Fig. 1g). Microglial process density in the top layer increased significantly by 21% ($p < 0.05$) as early as 2 h of incubation and by 29% ($p < 0.0001$) after 5 h of incubation (Fig. 1h, blue). Contrary to microglial cell bodies, process density at the bottom layer showed a significant drop by 15% ($p < 0.05$) already after 2 h of incubation (Fig. 1h, gold). Here, we also observed 23% ($p < 0.0001$) lower process density at the bottom region of slices compared to the top after 5 h of incubation (Fig. 1h; black statistical indication). We found that PSB application or lack of fractalkine signalling markedly inhibited slice-cutting-induced process redistribution (Fig. 1i). We also performed measurements on 450 μm-thick slices and found that slice thickness did not influence microglial translocation (Supplementary Fig. 1a).

To capture these changes in real-time, slice preparations were transferred into a recording chamber for confocal imaging (Fig. 1j). The native signal of microglia (CX3CR1[+/GFP]) was continuously imaged for at least 6 h after slice preparation (Fig. 1k, Supplementary Movies 1, 2). We found that individual cell behaviours correlated with the quantitative measurements above (Fig. 1e, h), with both processes and the cell body of microglia (Fig. 1k, coloured dots) expressing directed movement towards the top surface of slices (Fig. 1k, white dashed line).

Finally, we calculated the lowest distance of required cell displacement to reach the 5 h distribution starting from the 0-min distribution on the histological samples (Fig. 1l). Our results show that the translocation of cell bodies is a composite of two effects: a stronger effect acting in the direction of the top surface and a smaller effect acting towards the top and bottom cut surfaces away from the middle of the slices. Importantly, the total percentage of area covered by microglial processes dropped to half between the 0-min (acute slices fixed immediately after cutting) and 5-h timepoints (Mann–Whitney, $p < 0.0001$, Fig. 1l). Collectively, these results confirm that slice cutting procedure induces a strong migration of microglial cell bodies and processes towards the surfaces of acute slices via purinergic and fractalkine signalling pathways.

### Microglia undergo rapid, progressive morphological changes in acute slices in a P2Y12R- and CX3CR1-dependent manner

Microglial cells are well known to extend their processes towards injury as an early response, followed by marked phenotypic transformation[37–39]. The observed decrease in the area covered by microglial processes (Fig. 1l) also suggested robust microglial changes taking place in acute slice preparations. This behaviour was also evident in the data gathered by our imaging experiments, where we saw the rapid phenotypic transformation of individual microglial cells (Fig. 2a, Supplementary Movie 3). Since microglial phenotype changes have been shown to correlate with their resting membrane potential, we performed electrophysiological recordings from microglia in acute slices (Fig. 2b). Our results showed that microglial cells became gradually more depolarized throughout the total 5 h of the incubation process (Fig. 2c, top; $N = 158$ cells measured in slices from a total of 10 animals, one-way ANOVA with Dunnett's multiple comparison test), while we did not observe significant changes in input resistance (Fig. 2c, bottom).

We went on to define the spatio-temporal characteristics of the pronounced morphological changes. Preparations were immersion-fixed at different timepoints during incubation, while a group of mice

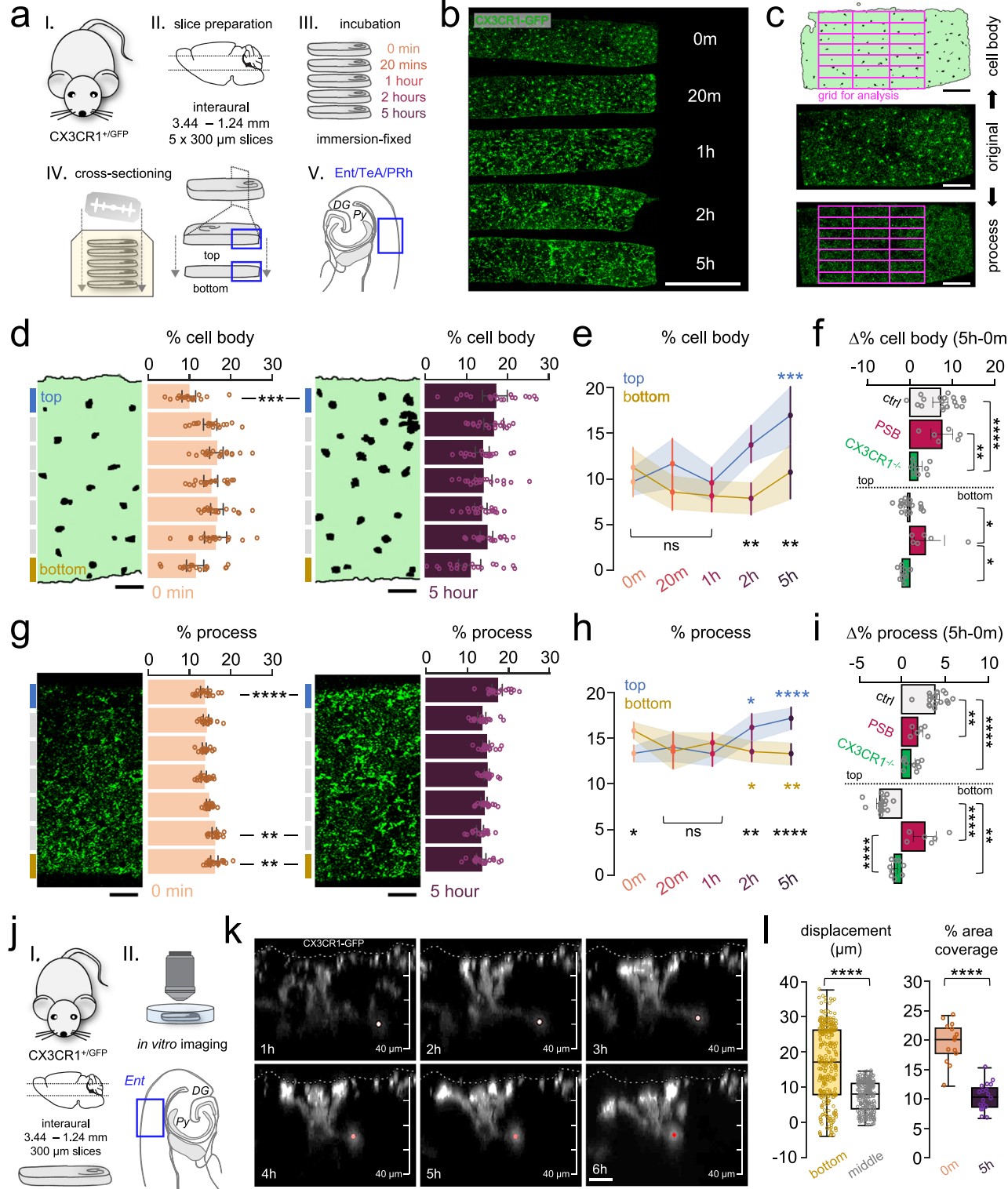

was transcardially perfused to obtain slice preparations from the same brain area as controls (Fig. 2d). Confocal images were analysed using unbiased automated morphological analysis[50] (Fig. 2e). Slice preparation induced drastic morphological differences in microglia. We observed rapid retraction of processes already visible in maximum intensity projection images created from the analysed z-stacks (Fig. 2f), as well as on individual cell morphologies and skeleton reconstructions via the analysis tool (Fig. 2g; Supplementary Movie 4). Sphericity values of individual cells (amoeboid shape and retraction of processes translate to higher sphericity values) peaked at ~100%

increase between 1 and 2 h of incubation both in regions of the cortex and in the hippocampus (Fig. 2h, one-way ANOVA with Dunnett's multiple comparison test, $p < 0.0001$). At the same time, total number of process endings dropped by ~50% after 20 min of incubation, and this reduction also peaked between 1- and 2 h of incubation with ~80% decrease in average (Fig. 2h, $p < 0.0001$). We also tested whether the observed changes would occur in older mice, by analysing the time-dependent morphological changes in slices prepared from 95 days old animals (Fig. 2i). We found that microglial phenotype changes tightly matched those observed in slices from younger animals. As both

**Fig. 1 | Microglia gradually migrate towards the surface of acute slice preparations in a P2Y12R- and CX3CR1-dependent manner. a** CX3CR1[+/GFP] littermates (males) were used to create slice preparations and immersion-fixed at different time-points, re-sectioned, and measured in Ent/TeA/PRh areas. **b** Cross-sections preserving top/bottom directionality. Scale: 500 μm (one independent experiment, $N = 3$ animal, $n = 3$ slices/animal). **c** A grid (violet) was used for quantification of cell body/process distributions. Scale: 100 μm. **d** Representative sections of images. Scale: 50 μm. Cell body % respective to grid layers. $N = 3$ animal, $n = 3$ slices/animal; P35 days; mean ± SEM, two-way ANOVA, Tukey's multiple comparison, $F(24, 408) = 2.305$, $p = 0.0005$. Source data are provided as a Source Data file. **e** Cell body distribution changes in top/bottom layers (blue/yellow). $N = 3$ animal, $n = 3$ slices/animal; P35 days; mean ± SEM. Blue stat.: compared to 0-min, black stat.: top/bottom means, two-way ANOVA, Tukey's multiple comparisons, $F(24, 408) = 2.305$, $p = 0.0005$. Source data are provided as a Source Data file. **f** Cell body distribution changes (control/PSB-treated/CX3CR1[−/−]). Ctrl: $N = 3$ animal, $n = 3$ slices/animal; PSB: $N = 2$ animal, $n = 3$ slices/animal; CX3CR1[−/−]: $N = 3$ animal,

$n = 3$ slices/animal; P35 days; mean ± SEM; two-way ANOVA, Tukey's multiple comparison, $F(2,60) = 13.65$, $p < 0.0001$. Source data are provided as a Source Data file. **g** Same as in **d** respective to processes, two-way ANOVA, Tukey's multiple comparisons, $F(24,408) = 4,766$, $p < 0.0001$. Source data are provided as a Source Data file. **h** Same as in **e** respective processes, two-way ANOVA, Tukey's multiple comparison test, $F(24,408) = 4,766$, $p < 0.0001$. Source data are provided as a Source Data file. **i** Same as in **f** regarding processes, two-way ANOVA, Tukey's multiple comparison tests, $F(2,60) = 17.23$, $p < 0.0001$. Source data are provided as a Source Data file. **j** CX3CR1[+/GFP] mice (males, P45–80 days) were used to create slices and measured via 2P imaging. **k** Supplementary Movie 1: translocation of cell body/processes (dot) towards the surface (stripped line). Scale: 10 μm. **l** Left: minimum required displacement (μm) towards the top measured from bottom/middle (gold/grey). Right: area covered by processes at 0-min/5-h (orange/purple). $N = 3$ animal, $n = 3$ slices/animal, P35 days; Mann–Whitney (two-sided), $p < 0.0001$. Boxes: interquartile range, whiskers: min–max, vertical bar: median. Source data are provided as a Source Data file.

purinergic and CX3CR1 signalling are heavily implicated in the regulation of microglial functions, we tested their possible involvement in slice-cutting-induced changes in microglial morphology. The lack of P2Y12R-s (in slices from P2Y12R[−/−] mice) markedly decreased the extent of time-dependent morphological changes after 5 h (Fig. 2j, left, two-way ANOVA with Tukey's multiple comparison test), while the lack of fractalkine-signalling in slices from CX3CR1[−/−] mice enhanced changes towards the 2-h time point (Fig. 2j, right), resulting in similar outcomes after 5 h. Based on these observations, we concluded that microglia show rapid, P2Y12R- and CX3CR1-dependent morphological changes in acute slice preparations both from young and older mice, and these changes are accompanied by the depolarization of microglial resting potential.

## Migration and rapid phenotype changes of microglia during the incubation process do not depend on cutting techniques

To investigate whether different slice-cutting techniques could influence the extent or temporal course of microglial phenotype changes during the incubation process, we next compared acute slice samples across three independent laboratories (lab #1, lab #2, lab #3) using identical fixation protocol and timing for serial preparations, but left native, well-established slice cutting techniques intact by each laboratory. Briefly: lab#1 used an ice-cold, sucrose-based cutting solution, and slices were incubated in an interface-type chamber filled with carbonated standard ACSF. Lab #2 used an ice-cold, N-methyl-D-glucamine (NMDG)-based cutting solution and slices were incubated in a low Na+, sucrose-based ACSF while floating in a Styrofoam "boat" with a netting at the bottom on the surface of an incubation chamber; lab #3 used an ice-cold, sucrose-based ACSF solution, and slices were incubated in a storage chamber filled with carbonated standard ACSF (for details see: Methods). Of note, we also tested slice preparation using a room-temperature cutting solution, which did not alter the course of microglial changes (Supplementary Fig. 1b). In all laboratories, slices were immersion-fixed in 4% PFA. Then, histological processing, imaging and analysis were performed in a uniform and blinded manner at the IEM (Fig. 3 a–d). Confirming our earlier results, both the sphericity of microglial cells (Fig. 3e) and the number of ending nodes (Fig. 3f) showed closely identical changes, similar to microglial cell body migration and process movements (Fig. 3g), suggesting that the observed changes are general, common characteristics of different acute slice preparation techniques. These results confirm that slice-cutting-induced migration and rapid phenotype changes of microglia during the incubation process do not depend on cutting techniques. As microglia have critically important roles during development that can differ substantially from those of adult microglia, and the acute slice model is also widely used in developmental studies, we have repeated some experiments on slices from postnatal day 18–20 (P18–20) mice. We found that time-dependent microglial cell body and

process translocation (Supplementary Fig. 1c, and also time-dependent changes of microglial morphology (Supplementary Fig. 1d) happen in slices from mice in the late postnatal developmental phase in a very similar manner as in adulthood, underscoring the broad impact of the cutting-induced changes we observed (see also averages of individual animals in Supplementary Fig. 2).

## Slice cutting induced rise in extracellular ATP followed by sustained induction of focal ATP events characterize acute brain slices

Because microglial phenotype changes in acute slices were dependent on purinergic signalling, while injury-induced ATP is known to markedly affect microglial phenotypes and recruitment via P2Y12R[37,45], we set out to examine changes in extracellular ATP levels in acute brain slices. We generated a novel mouse strain (ATP1.0[ΔVglut1]) by expressing a sensitive, genetically encoded fluorescent ATP sensor (GRAB$_{ATP}$) in VGluT1 positive neurons[51,52]. In order to capture time-dependent changes in extracellular ATP levels, we recorded 10 min long videos with two-photon imaging, followed by confocal z-stacks from both regions of cortex (entorhinal, temporal association, perirhinal; Ent, TeA, PRh) and hippocampus (CA 2-3 str. radiatum), repeated once in every hour after slice preparation (Fig. 4a). We observed high fluorescence signal, representing increased extracellular ATP levels both in the hippocampal CA3 region and in cortical layers shortly after slice preparation (Fig. 4b), which was most intense near the slice surface, as revealed by Z stack acquisitions. The mean ATP signal rapidly declined during the first 10 min of imaging immediately after slice preparation (Fig. 4b, c; Supplementary Fig. 3a; Supplementary Movie 5). Then on, ATP levels showed a slow, gradual decrease (Fig. 4d left and f; Supplementary Fig. 3a, b) with ATP gradient maintained through the 0–100 μm depth of the tissue for at least 5 h (Supplementary Fig. 3c). Ambient ATP signals corresponded to an injury-related rise in extracellular ATP due to the slice preparation process, as suggested by the induction of ATP signals at sites of secondary, mechanical injury (Fig. 4e and Supplementary Movie 6) caused immediately prior to the start of imaging even several hours after slice preparation (see also Supplementary Fig. 4a). Importantly, the immediate, intense ATP response due to secondary injuries and a subsequent, continuous decay were both prevalent in older (P74 days) and late postnatal (P20 days) animals. Time-dependent decline of focal ATP signals indicated functional ecto-ATPase activity in acute brain slices, which we confirmed via NTPDase inhibitor (ARL67156, 10 μM, Supplementary Fig. 4b, c). Intense, focal events of ATP release were observed throughout the 5 h imaging period in response to slice preparation (Supplementary Movie 5; Fig. 4d right). The average ATP event incidence in acute slices was 12.3 ± 2.2 (hippocampus) and 25.4 ± 5.2 (cortex) during the 10 min imaging periods, resembling systemic inflammation-induced flashing activity in vivo[52] (Fig. 4g,

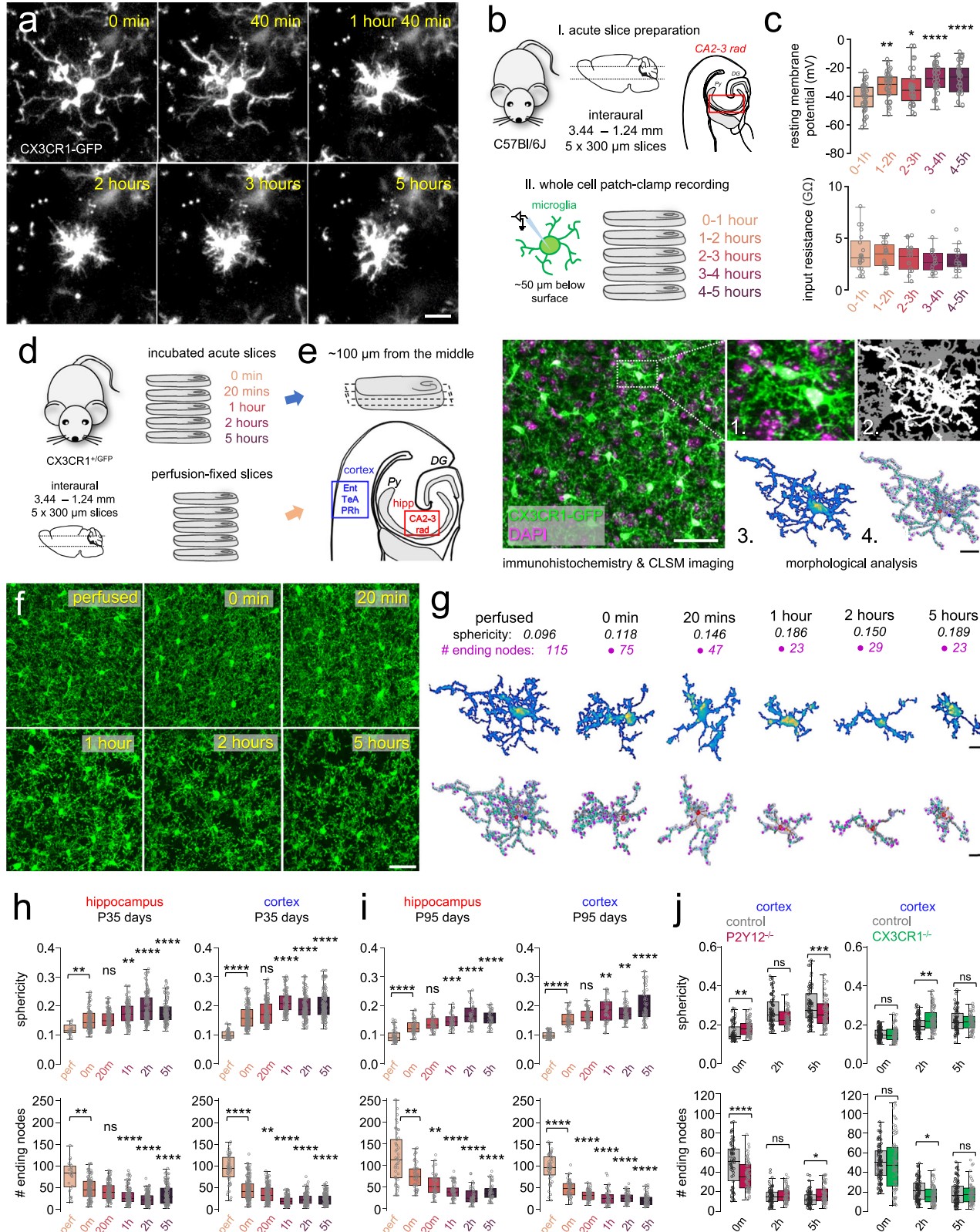

Supplementary Fig. 5a). While we detected more frequent and intense ATP events in cortical regions of acute slices than in the hippocampus (Fig. 4g, h), all other characteristics (ATP event area, duration, rise time, decay time) were quite similar in both regions (Supplementary Fig. 5b).

We observed that the ATP event parameters (amplitude, duration, area) showed a log-normal distribution with a high degree of variance,

therefore we hypothesized that these events might correspond to different mechanisms or sources of ATP release. *K*-means cluster analysis of the ATP event parameters revealed the existence of at least two separate groups both in the hippocampus and cortex, differentiating a dimmer/faster/smaller (flash) and a significantly brighter/slower/bigger (surge) subset of events (Fig. 4i, j). Flash intensities were usually around or below the 100 nM ATP concentration range, as

**Fig. 2 | Microglia undergo rapid, progressive morphological changes in acute slices in a P2Y12R- and CX3CR1-dependent manner. a** Supplementary Movie 3. CX3CR1$^{+/GFP}$ signal of microglia after slice preparation. Scale: 10 μm. **b** Acute slices obtained from CX3CR1$^{+/GFP}$ (male littermates). Microglia were measured in whole cell patch-clamp, targeted -50 μm below the surface (CA2-3 stratum radiatum). **c** Resting membrane potential(top)/input resistance(bottom) during recovery. $N = 10$ animal (7 males, 3 females), $n = 5$ slices/animal, P90 days; $N = 158$ cells, one-way ANOVA, Dunnett's multiple comparisons, $F(4,150) = 9.768$, $p < 0.0001$ (top), ns (bottom). Boxes: interquartile range, whiskers: min–max, vertical bar: median. Source data are provided as a Source Data file. **d** Acute slices were obtained from CX3CR1$^{+/GFP}$ (male littermates) and immersion-fixed after different time-points (blue arrow), or slices were obtained from perfusion-fixed animals (orange arrow). **e** Slices were re-sectioned (middle 100 μm), and measurements were performed in Ent/TeA/PRh and CA2-3 stratum radiatum. Stained images (left, Scale: 20 μm) were analysed: raw z-stack (1.), image segmentation (2.), cell segmentation (3.), cell body(yellow)/processes(blue). Obtaining skeleton (4.) 3D-models (right, bar:

10 μm) (two independent experiments, $N = 4$–4 animal, $n = 3$ slices/animal). **f** Analysed confocal images (Scale: 50 μm). **g** Top: individual microglia (yellow: cell body; blue, purple: processes; Scale: 5 μm). Bottom: skeletons (violet, cell body: red dot, Scale: 5 μm). **h** Top: sphericity in cortex (blue) and in hippocampus (red). Bottom: # process endings/cell. $N = 8$ animals, P35 days, one-way ANOVA, Dunnett's multiple comparison, $p < 0.0001$ (comparisons from 20 min/5 h are made to 0 min) Boxes: interquartile range, whiskers: min–max, vertical bar: median. All $p$ and $F$ values can be found in Supplementary Table 2. Source data are provided as a Source Data file. **i** Same as in **h** from P95 animals. $N = 5$ animals, P95 days, one-way ANOVA, Dunnett's multiple comparisons, $p < 0.0001$. Source data are provided as a Source Data file. **j** Sphericity/# process endings in CX3CR1$^{-/-}$ (green) or P2Y12R$^{-/-}$ (purple) and WT (grey), $N = 3$ animal/condition (only males), $n = 5$ slices/animal, P35 days, two-way ANOVA with Tukey's multiple comparison test, $p < 0.0001$. Boxes: interquartile range, whiskers: min–max, vertical bar: mean. All $p$ and F values can be found in Supplementary Table 2. Source data are provided as a Source Data file.

---

determined by local ATP puffs defining sensor linearity (Fig. 4i and Supplementary Fig. 4d), while surges were usually between the 100 nM–1 μM ATP range. Importantly, the vast majority of the ATP events did not exceed -1 μM concentration and we have never detected persistent ATP events, indicating that ATP is released from operational cellular sources and subjected to effective degradation upon release. Interestingly, the prevalence of the larger/slower/brighter ATP surges largely increased after the first hour of slice preparation, and gradually decreased towards the 5th hour of the recovery period (Fig. 4k).

### Focal ATP events recruit microglial processes and shape microglial morphology in a P2Y12R- and CX3CR1-dependent manner

To understand how these ATP events regulate microglial behaviour, we injected AAVs to express ATP sensors in neurons of the cortex of CX3CR1-cre/tdTomato (microglia reporter) mice (Fig. 5a) and performed ex vivo two-photon imaging experiments (see the "Methods"). As above, $k$-means cluster analysis of the ATP events separated the same flash/surge subset of groups (Fig. 5b), even in the presence of the P2Y12R blocker PSB0739 (4 μM) or the CX3CR1 inhibitor AZD8797 (10 μM). ATP surges were more intense in slices subjected to CX3CR1 blockade compared to control and PSB conditions, while CX3CR1 blockade resulted in shorter, and P2Y12R blockade in longer duration times of ATP surges compared to control (Fig. 5c).

Importantly, microglial processes were rapidly (within 1–3 min) recruited to spontaneously emerging focal ATP events with a directed movement (dm) towards the centre of flashes/surges (Supplementary Movie 7, Fig. 5d, e). We quantified these microglial actions as the change in the tdTomato signal (Δ% MFI, see the "Methods" section) measured in the area (ROI) of flashes (Supplementary Fig. 4e) and surges (Fig. 5e) before and after a given ATP release event. Of note, blockade of either P2Y12R (PSB) or CX3CR1 receptors (AZD) largely prevented microglia process recruitment to focal ATP events categorized as flashes (Fig. 5f, left), while directed movements towards surges seemed to be less affected by AZD treatment. No effect on movement speed (quantified as the latency of initiated movements) was seen compared to control (Fig. 5f), indicating that P2Y12 and CX3CR1 blockade can influence specifically directed microglial movements. In line with this, we observed that while the majority (72%) of ATP events were followed by directed movement (Fig. 5g), blocking P2Y12 receptors largely (to 27%) and CX3CR1 receptors moderately (to 52%) reduced the prevalence of directed microglia process movement (Fig. 5g). In control slices, cells responding to ATP events even 5 h after slice preparation were observed with a decreasing trend of directed movements over time, which remained constantly low in the presence of PSB or AZD. Microglia movement to ATP flashes was also affected by P2Y12R and AZD, whereas movements to ATP surge events were only affected by PSB (Fig. 5h). Altogether, our data indicate, that ATP

activity and in particular, directed microglial process recruitment to focal ATP events take place via P2Y12R and CX3CR1, contributing to microglial phenotype changes in acute brain slices.

### Microglial P2Y12R undergoes rapid downregulation during the incubation process, paralleled by changes in microglia-neuron interactions

P2Y12R is a core microglial receptor through which microglia sense and influence neuronal activity and fate[53–57], and which is downregulated upon injury and inflammatory challenges[37,58]. Interestingly, pre-embedding immunofluorescent labelling indicated a marked reduction of P2Y12R labelling intensity already 1 hour after slice preparation (Fig. 6a). To quantitatively examine these changes during the whole incubation process, acute slice preparations were immersion-fixed at different timepoints during incubation (Fig. 6b), followed by visualization of P2Y12R with a recently developed quantitative post-embedding immunofluorescent labelling technique[59], enabling unbiased assessment of P2Y12R labelling intensity changes (Fig. 6c). As each ultrathin section represents the whole cross-section of one acute slice (Fig. 6c/4), the measurement could be performed throughout the whole depth range of acute slices within a single image plane. We quantified P2Y12R labelling intensity at microglial cell bodies (Fig. 6d, left), thick processes (Fig. 6d, middle) and thin processes (Fig. 6d, right), to investigate whether dedicated subcellular compartments are differentially affected. We performed the measurements excluding the top and bottom 40 μm at the surfaces. Confirming our immunofluorescent results, we found a consistent, rapid decrease in P2Y12R labelling intensity in all the compartments until 1 h of incubation, followed by a slight increase towards the 5-h timepoint (Fig. 6e). Loss of P2Y12R was most prominent in thick and thin processes (Fig. 6e, middle, right), and less apparent on microglial cell bodies (Fig. 6e, left). As such, already after 1 h of incubation, the labelling intensity of P2Y12R (arbitrary unit, see the "Methods" section) dropped by 25% on thick processes ($p < 0.0001$), 29% on thin processes ($p < 0.0001$) and 20% ($p < 0.05$) at cell bodies of microglia. We did not find any gradient along the depth within single acute slices regarding P2Y12R downregulation (Supplementary Fig. 6). These results confirm a robust downregulation of the core microglial purinergic receptor, but—in spite of this decrease—there is still sufficient receptor expression on microglial membranes to enable an effective and functional P2Y12R-signalling even 5 h after slice cutting, evidenced by the functional experiments using PSB.

As P2Y12R-signalling is critically important for somatic junctions (interactions between microglial processes and neuronal cell bodies[53]), we hypothesized that the contact prevalence and coverage of neuronal cell bodies could be affected. To assess possible changes regarding somatic junctions, we enhanced the native GFP-signal of microglia by anti-GFP labelling, while neuronal cell bodies were identified via Kv2.1 labelling (Fig. 6f). We found that the total percentage of neuronal soma

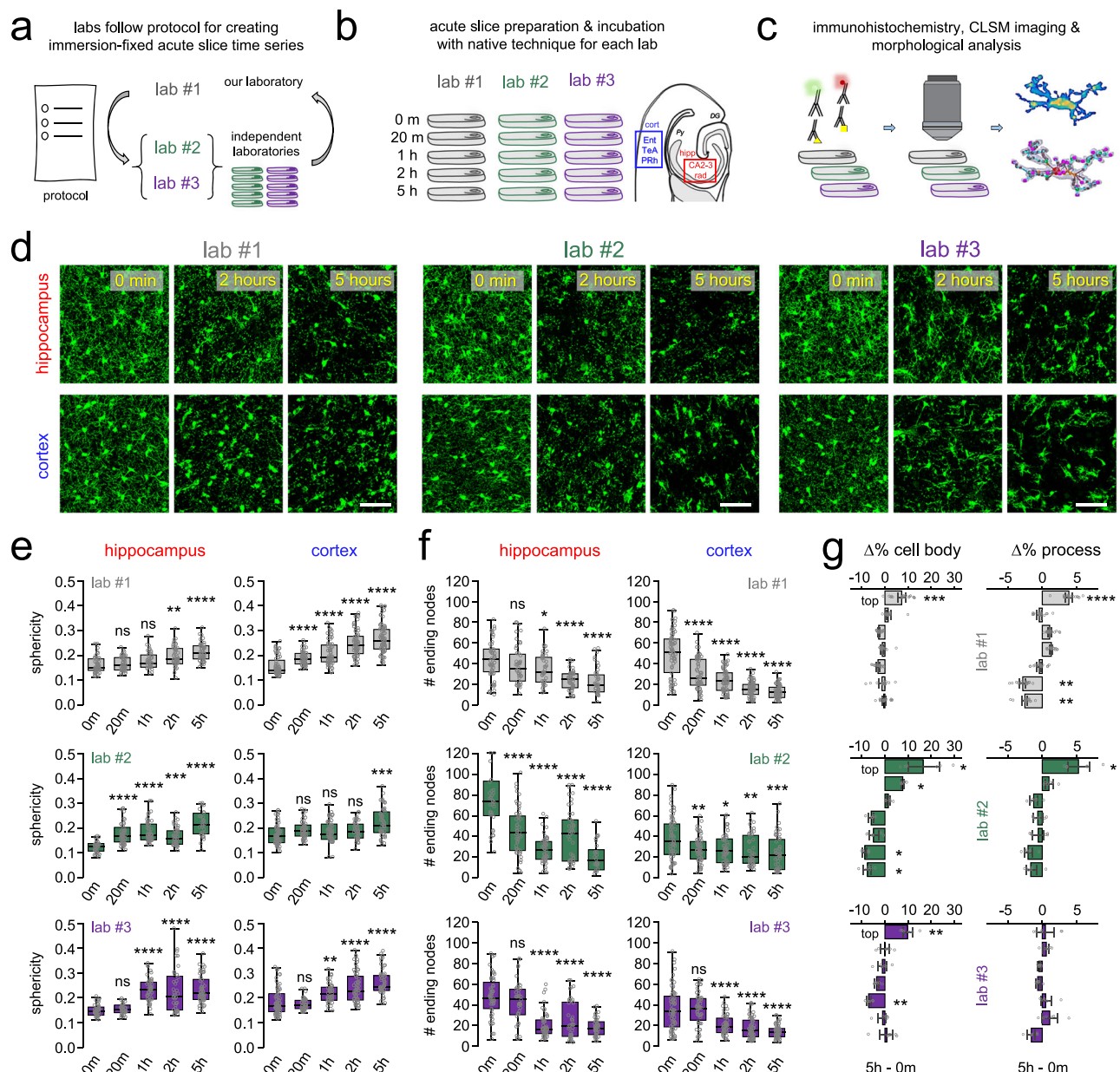

**Fig. 3 | Migration and rapid phenotype changes of microglia during the incubation process do not depend on cutting techniques. a** Schematics of experiment. Comparison of slice preparation techniques of independent laboratories. Laboratories received a protocol to synchronize acute slice fixation timepoints and fixation methods, while using their own native acute slice preparation method. **b** Measurements were performed in Ent/TeA/PRh cortical areas, and CA2-3 stratum radiatum of the hippocampus from C57Bl/6J animals (only males). **c** Acute slice preparations were sent to our laboratory after fixation and preparation for transport (see the "Methods" section), where they were treated together during immunostaining, imaging, and morphological analysis. **d** Maximum intensity projections of confocal images showing microglia (IBA-1) in the hippocampus (top, red) and in the cortex (bottom, blue). Scale bar: 50 μm. (One independent experiment ($N$ = 3-2-2 animal, $n$ = 1 slice/animal/timepoint). **e** Quantification of extracted morphological features across different laboratories regarding sphericity in hippocampus (left) and in cortex (right). $N$ = 3 animal/lab1; 2 animal/lab#2; 2 animal/lab#3; $n$ = 3 slices/animal, P65 days, one-way ANOVA with Dunnett's multiple comparison test, $p < 0.0001$. Boxes: interquartile range, whiskers: min–max, vertical bar: mean. All $p$ and $F$ values can be found in Supplementary Table 3. Source data are provided as a Source Data file. **f** Same as in **d**, regarding number of ending nodes/cell in cortex (left) and in hippocampus (right). $N$ = 3 animal/lab1; 2 animal/lab#2; 2 animal/lab#3; $n$ = 3 slices/animal, P65 days, median ± SEM, one-way ANOVA with Dunnett's multiple comparison test, $p < 0.0001$. Boxes: interquartile range, whiskers: min–max, vertical bar: mean. All $p$ and $F$ values can be found in Supplementary Table 3. Source data are provided as a Source Data file. **g** Bar plots representing measured changes of microglial cell body numbers (left) and area covered by processes (right) in percentages calculated between 0 min and 5 h across the top and bottom layers of acute slice preparations. $N$ = 3 animal/lab1; 2 animal/lab#2; 2 animal/lab#3; $n$ = 3 slices/animal, P65 days, median ± SEM, two-way ANOVA with Dunnett's multiple comparison test, $p < 0.0001$, mean ± SEM. All $p$ and $F$ values can be found in Supplementary Table 3. Source data are provided as a Source Data file.

contacted by microglial processes gradually decreased throughout the incubation process and dropped by ~23% (one-way ANOVA, $p < 0.0001$) after 5 h of incubation (Fig. 6f, left). We also investigated the percentage of somatic area covered by microglial processes over

time, which has doubled at 0 min (fixed immediately after cutting) compared to the perfused controls (from 4% to 8%, $p < 0.01$). This observation indicates rapid microglia actions taking place during the ~1–3 min between brain extraction and slice fixation. The massive

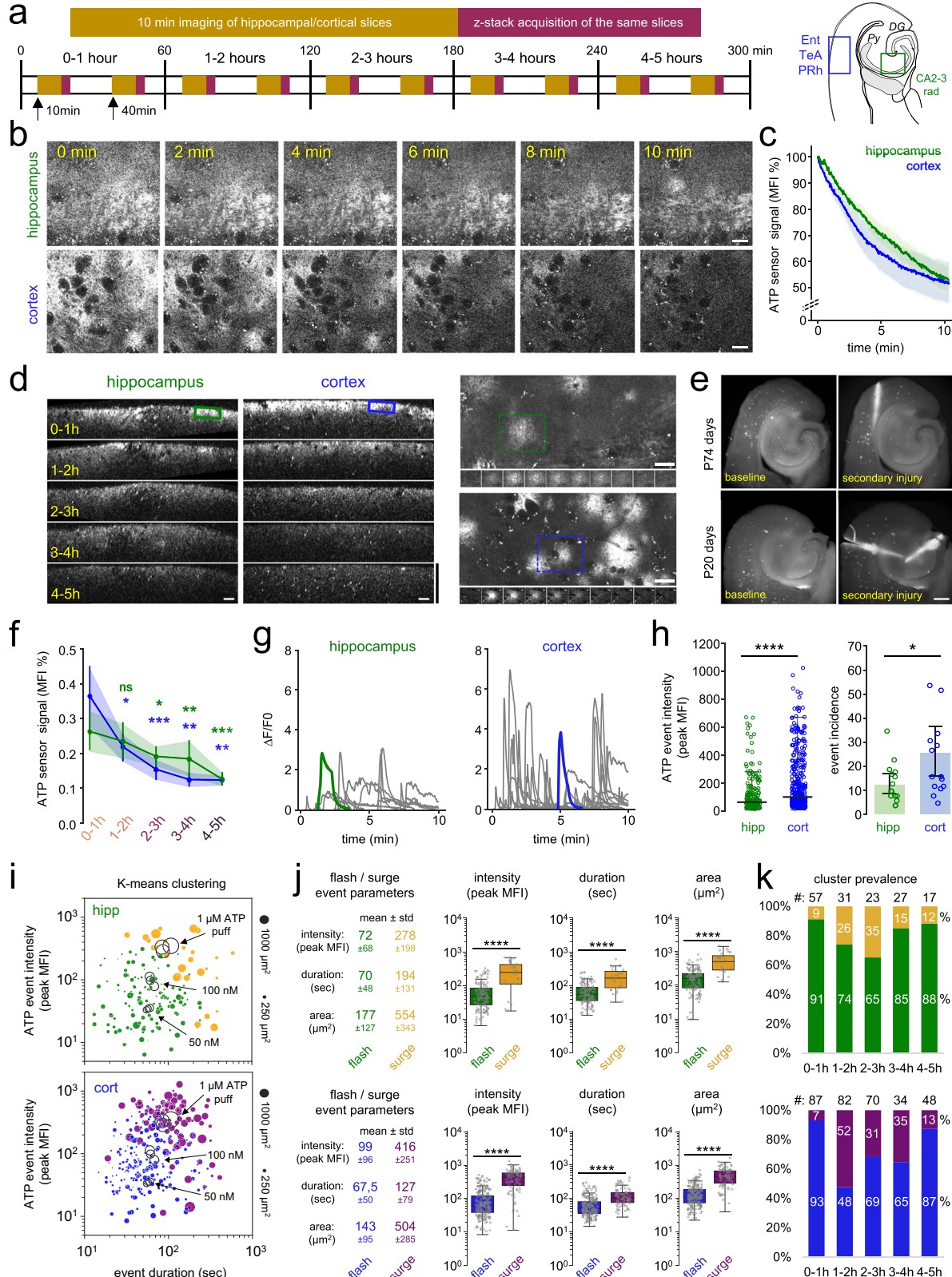

initial increase was followed by a gradual decrease dropping back again to control values after 5 h of incubation (Fig. 6g, right).

### Rapid changes in microglia-synapse interactions and microglia-dependent synaptic sprouting characterize acute slices

Although P2Y12R-signalling is critically important for somatic junctions, this pathway seems to be less involved in regulating microglia-synapse interactions[53]. Nevertheless, the observed profound changes in microglial phenotype and the sustained and complex ATP events could still lead to substantial alterations in microglia-synapse interactions after acute slice preparation. To assess whether these time-dependent changes in microglial phenotype would indeed parallel altered microglia-synapse interactions, we also looked at microglial contacts with synaptic elements[53]. Synapses were

**Fig. 4 | Slice cutting induced rise in extracellular ATP followed by sustained induction of focal ATP events characterize acute brain slices. a** Ex-vivo 2P imaging of the ATP sensor (GRABATP) in VGluT1 neurons, followed by z-stack acquisition. ($N = 3$ animals, only males, P70 days). **b** Representative images of the videos after cutting. Scale: 20 μm. **c** MFI data showing a continuous decrease of extracellular ATP signal. Two-way RM ANOVA, Time: $F_{(1.169, 21.04)} = 67.03$, $p < 0.0001$. Regions: $F_{(1, 18)} = 0,3801$, $p = 0,5452$, mean ± SEM. Source data are provided as a Source Data file. **d** Side-views of Z-stacks: elevated ATP at the slice surface (0–25 μm, left). Horizontal scale: 20 μm, vertical scale: 125 μm. Occurrence of ATP flashes (right), timeline of two ATP events (bottom, 10 s/image). Scale: 20 μm. **e** MIP images from 10-min imaging before/after (1–2 min) deliberate incisions with sterile blades (controlled secondary injury), ~4 h after cutting, resulting in elevated ATP. Scale: 500 μm. **f** Quantification of (**d**), ATP levels follow a time-dependent decrease (5-h vs. to 0-1 h). Two-way RM ANOVA, Tukey's multiple comparison, $F_{(4,64)} = 8.267$, $p < 0.0001$, mean ± SEM ($N = 3$ animals, $n = 3$ slices, 4 ROIs/region) Source data are provided as a Source Data file. **g** $\Delta F/F$ graphs depict ATP event activity as shown in (**d**) and (**e**). Source data are provided as a Source Data file. **h** Intensity/incidence of ATP events in cortex ($n = 360$) vs. hippocampus ($n = 181$). Number of ATP events/10-min imaging/slice. Mann–Whitney (two-sided), $p < 0.0001$, median ± SD (left); unpaired $t$-test (two-sided), $p = 0.0258$, mean ± SEM (right). Source data are provided as a Source Data file. **i** $k$-means clustering reveals two different event clusters. Size of dots reflect size of flashes/surges (black indication). See also Supplementary Fig. 4d ($N = 3$ animals, $n = 10$ slices) Source data are provided as a Source Data file. **j** ATP flash/surge characteristics, Mann–Whitney (two-sided), $p < 0.0001$ ($N = 3$ animals, $n = 10$ slices, 5–5 slices/timepoint). Boxes: interquartile range, whiskers: min–max, vertical bar: median. Source data are provided as a Source Data file. **k** Cluster prevalence changes over time after slice preparation. Note: substantial increase of ATP surge prevalence 1–2 h after slice cut ($N = 3$ animals, $n = 10$ slices, 5–5 slices/timepoint). Source data are provided as a Source Data file.

identified via VGLUT1-Homer1 and VGAT-Gephyrin co-localization (Fig. 7a). Microglia–synapse contact prevalence showed similar alterations after slice preparation in the case of glutamatergic synapses (Fig. 7b, left), where average percentage of contacts increased by 46% between the perfused and 0-min conditions ($p < 0.01$) and gradually dropped back to control values. In the case of GABAergic synapses (Fig. 7b, right), we did not observe a significant increase immediately after slice preparation, however, we measured a prominent decrease in contacts during the incubation process (compared to 0 min), with a 33% drop already after 20 min of incubation ($p < 0.01$), which further decreased to a 47% drop after 5 h. Based on these observations, we concluded that microglia–neuron interaction sites underwent rapid and progressive changes, as somatic coverage and contact prevalence on glutamatergic synapses significantly increased immediately after slice preparation, followed by a gradual decrease over time.

Given the observed changes in microglial contact prevalence on synapses, we next examined how the density of synaptic elements changed during the incubation process. We used the quantitative post-embedding labelling method to precisely determine both glutamatergic and GABAergic synaptic density changes (Fig. 7c) in acute slices. We found significant increases in both glutamatergic (20% increase after 1 h, $p < 0.05$) and GABAergic (20% increase after 2 h, $p < 0.01$) synaptic density (Fig. 7d), similarly to previous reports[60–62]. While we observed similar, gradual increase in glutamatergic and GABAergic synaptic density towards the 1- and 2-h timepoints, there was a 1-h lag in the case of GABAergic synapses (Fig. 7d, left vs. right), which peaked after 2 h of incubation. We observed the highest density of glutamatergic synapses after 5 h (28% increase compared to 0 min, $p < 0.01$). Based on these observations, we concluded that the neuronal network is likely to be reorganized by the slow and gradual build-up of excitatory synapse numbers, which is followed by the inhibitory synapses in the same manner. The inhibitory sprouting seemed to reach its maximum after 2 h of incubation, while excitatory synapse numbers further increased towards the 5-h timepoint.

Increasing evidence suggests that microglial cells are essential for the formation, pruning and maintenance of synapses both in the developing and in the adult CNS[63]. We wanted to examine whether microglia can also actively contribute to the observed time-dependent structural and functional changes in synaptic density detected in acute slice preparations. To this end, we measured changes in synaptic densities in acute slice preparations in the absence of microglia (induced by elimination of microglia with PLX5622 for three weeks in vivo prior to slice preparation, Fig. 7e, f). Depletion of microglia with PLX3373 or PLX5622 in vivo has previously been shown to induce no substantial changes in neuronal cell numbers and morphology, nor did it cause any deficits in cognition and behaviour, although dendritic spine numbers showed a slight increase[64–66]. To measure and compare synaptic density changes between control (CTRL) and depleted (DEPL) conditions, we used the previously described post-embedding

labelling method (Fig. 6c). We found that time-dependent synaptic density changes after slice preparation were markedly influenced by microglial actions (Fig. 7g). To our surprise, the absence of microglia abolished the gradual increase of glutamatergic synaptic density observed under control conditions and resulted in significantly lower synapse densities from 20 min after slice preparation and onward, reaching 27% lower synaptic density values in DEPL compared to CTRL after 5 h ($p < 0.0001$, Fig. 7h, left).

Even more prominent differences were observed regarding GABAergic synaptic densities (Fig. 7h, right). The absence of microglia led to a marked initial increase in GABAergic synaptic density, peaking at 1 h, followed by a significant drop at 2 h. This course fundamentally differs from that observed under control conditions, which peaks at 2 h (19% higher in CTRL, $p < 0.001$, Fig. 7h, right). These results indicate that microglia differentially control excitatory and inhibitory synaptic sprouting in acute slice preparations, initiating glutamatergic while repressing GABAergic synapse formation during the early stages (<1 h) following acute slicing, while microglial interactions with neuronal cell bodies also undergo robust, time-dependent changes.

## The absence of microglia, microglial P2Y12R or CX3CR1 function dysregulates sharp wave-ripple activity

The sharp wave-ripple (SWR) complex is a widely studied synchronous activity arising in the CA3 region of the hippocampus, which can spontaneously occur in acute slice preparations[6,47], and it is commonly associated with memory formation, consolidation, and planning[67,68]. It is well known that slices need a recovery period after cutting, to enable reliable and stable electrophysiological measurements, including SWR activity. We hypothesized that the observed profound changes of microglia-neuron interactions—regarding somatic coverage, somatic and synaptic contact prevalence and synaptic densities—could at least in part be responsible for slice recovery or slice maturation required for the emergence of network activity. Indeed, we observed a significant increase in SWR parameters after 2 h in control slices (Supplementary Fig. 7), which paralleled marked, time-dependent changes in GABAergic and glutamatergic synapse densities.

Because the absence of microglia resulted in markedly lower glutamatergic and GABAergic synaptic densities after 2 h of incubation (Fig. 7h), we set out to examine whether synchronous events are affected by microglial function. To this end, we compared features of spontaneously occurring SWR activity measured from control (CTRL), microglia depleted (DEPL), P2Y12 KO (P2Y12$^{-/-}$) or CX3CR1 KO (CX3CR1$^{-/-}$) acute slice preparations (Fig. 8a). Simultaneous recordings from control and depleted/KO conditions allowed precise pairwise quantification and comparison of the SWR activity between control and loss of function conditions. We observed that SWR prevalence (% of slices that generated spontaneous SWR activity) in control and CX3CR1$^{-/-}$ slices was around ~50% in all cases, while in depleted and P2Y12$^{-/-}$ slices the prevalence dropped dramatically (11% in depleted,

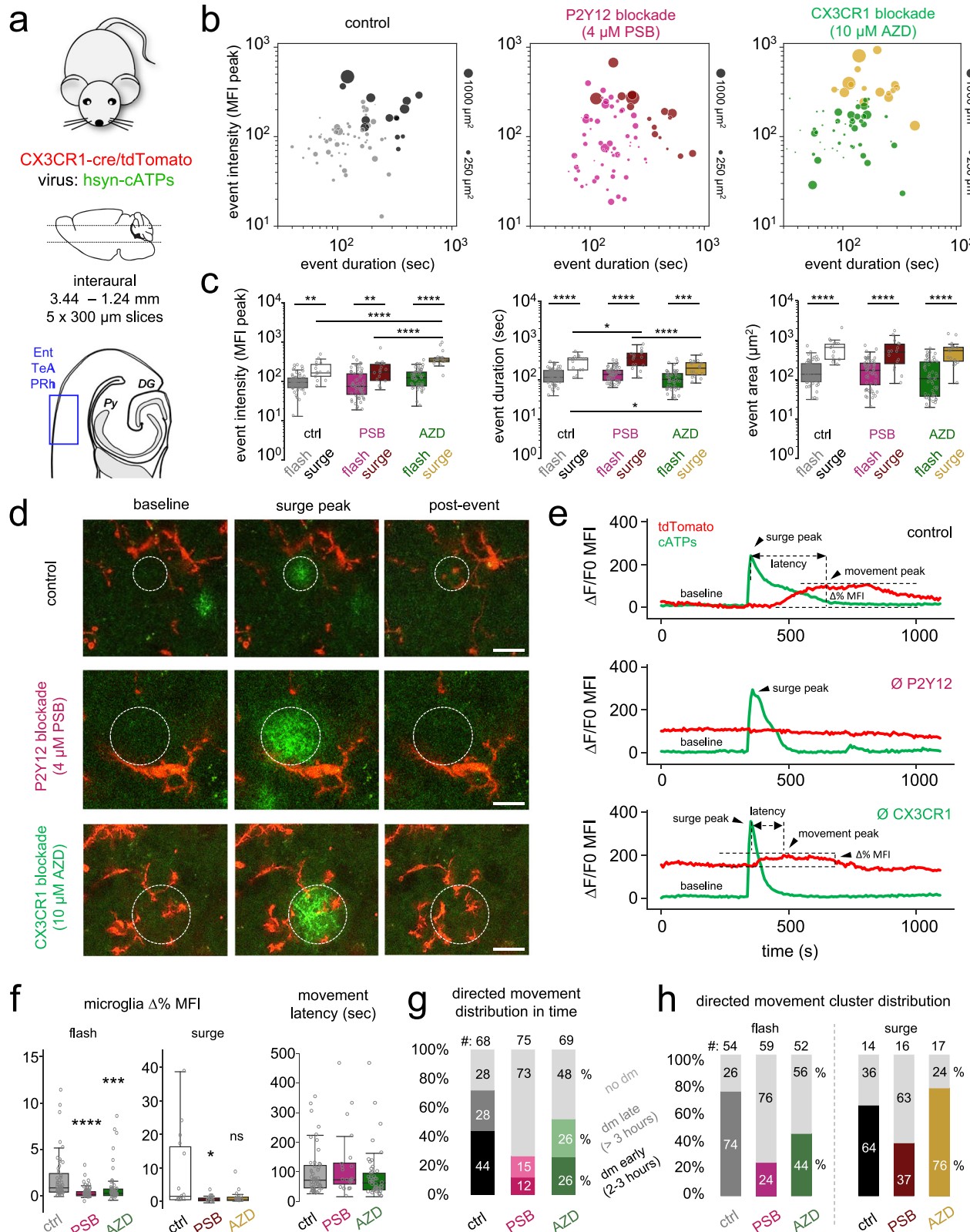

33.3% in P2Y12$^{-/-}$, Fig. 8e). This striking difference confirmed that microglial actions and P2Y12R functions are necessary for the emergence of physiological-like network activity in ex vivo slices. Furthermore, our results also showed significant differences between the spontaneously occurring SWR activity parameters (Fig. 8f). We observed that the amplitude of SWRs was significantly impaired by microglia manipulation (ctrl vs. depleted: ~4,6-fold $p < 0.0001$; ctrl vs.

P2Y12$^{-/-}$: ~1.7-fold, $p < 0.05$; ctrl vs. CX2CR1$^{-/-}$: ~1,5-fold, $p < 0.05$). We only observed significant difference in SWR rate when comparing control with depleted slices (ctrl vs. depleted: ~1.6-fold, $p < 0.05$), while the amplitude of ripple frequency was significantly impaired in all comparisons (ctrl vs. depleted: ~2.1-fold, $p < 0.001$; ctrl vs. P2Y12$^{-/-}$: ~1.3-fold, $p < 0.001$; ctrl vs. CX2CR1$^{-/-}$: ~1.2-fold, $p < 0.001$; Mann-Whitney test). Taken together, these results suggest that the

**Fig. 5 | Focal ATP events drive microglial process motility in acute brain slices in a P2Y12R- and CX3CR1-dependent manner. a** AAV-induced neuronal expression of ATP sensors (Cx3cr1-cre/tdTomato mice, $N = 9$ animals, only males, P70–110 days). Control ($n = 6$ slices), PSB0739-treated ($4\,\mu M$; $n = 8$ slices) or AZD8797-treated ($n = 5$ slices). **b** k-means clustering. Dot sizes represent the area of events (scale in right) Grey: control, pink: PSB, green: AZD show flashes, black: control; purple: PSB; yellow: AZD show surges ($N = 9$ animals, $n = 6$ ctrl, 8 PSB, 5 AZD treatment). Source data are provided as a Source Data file. **c** Comparison of ATP flash/surge characteristics of control, PSB0739 and AZD8797 treatment. Two-way ANOVA, Tukey's multiple comparisons, Intensity: $F_{(1, 206)} = 92.31$, Duration: $F_{(1, 206)} = 133$, Area: $F_{(1, 206)} = 115.8$, $p < 0.0001$ ($N = 9$ animals, $n = 6$ ctrl, 8 PSB, 5 AZD-treated slices). Boxes: interquartile range, whiskers: min–max, vertical bar: mean. Source data are provided as a Source Data file. **d** ATP events (green) and process displacement (red, Supplementary Movie 7). ROIs indicate ATP event areas at maximum intensity. Scale bars: $10\,\mu m$. **e** $\Delta F/F$ traces of ATP sensor activity (green)

and superimposed process accumulation (red) within the ATP flash territories, demarcated in **d**). **f** Left: Microglia process recruitment to flashes/surges represented as tdTomato $\Delta\%MFI$. Right: Latencies of process movements ($N = 9$ animals, $n = 6$ ctrl, 8 PSB, 5 AZD-treated slice), measured as indicated on (**e**). Kruskal–Wallis test with Dunn's multiple comparison test, $\Delta\%MFI$: $p < 0.0001$ (left), Latency: $p = 0.19$ (right). Boxes: interquartile range, whiskers: min–max, vertical bar: median. Source data are provided as a Source Data file. **g** Directed movements (dm) early (2–3 h) vs. late (3–6 h) stages after slice preparation ($N = 9$ animals, $n = 6$ ctrl, 8 PSB, 5 AZD treated slices). Source data are provided as a Source Data file. **h** Distribution of process movement prevalence in response to flashes/surges (9 animals, 6 ctrl, 8 PSB, 5 AZD treated slice). Light colours (grey: control; pink: PSB; green: AZD) represent movement towards flashes, darker colours (black: control; purple: PSB; yellow: AZD) represent movement towards surges ($N = 9$ animals, $n = 6$ ctrl, 8 PSB, 5 AZD-treated slices). Source data are provided as a Source Data file.

positive influence of microglia on SWR activity observed in acute slice preparations is in part mediated by microglial P2Y12R- and fractalkine-signalling, and that specific microglial actions are needed for the maturation of ex vivo acute slices.

To investigate the relevance of these observations in vivo, we performed chronic recordings after implanting 64-channel probes into the hippocampus of freely behaving mice in their home cage environment (Fig. 8g). After this, the recording was repeated on the same animals following microglia depletion. We found that while ripple frequency, length and amplitude did not change, the absence of microglia led to a significant decrease in ripple rate (Fig. 8h, i). These data suggest that microglia are important contributors to maintaining complex neuronal network activity patterns in vivo.

## Discussion

In this study, we provide a comprehensive assessment of microglial function in acute slice preparations in an experimentally relevant timeframe, using sample sets from multiple expert laboratories and advanced tools, including a novel mouse strain expressing a recently developed, highly sensitive fluorescent ATP sensor[52,69]. We show that while microglia are undergoing marked time-dependent phenotype changes, they remain instrumental in maintaining ex vivo spontaneous activity of the neuronal network in a P2Y12R- and CX3CR1-dependent manner. We believe that these mechanisms of microglia-neuron interactions, as influenced by the identified purinergic activity patterns, should be considered for studies on microglial and neuronal functions ex vivo while also having important implications for the acute slice methodology and could facilitate improvements in modelling. Finally, our findings on injury related ATP events or microglial influence on SWR activity both ex vivo and in vivo may have important implications for common neurological disorders.

We first studied, how the slice preparation procedure affected microglial cells over time. While locomotor and process polarization behaviour of microglia have been previously suggested in acute slices[38,39], population-level changes in the distribution of cell bodies and processes along the full depth of slices have not been investigated in a time-dependent manner before. The acute slice preparation technique can ultimately be considered as a traumatic event assumed to influence microglial function, as observed in acute slices prepared via different methodologies[10,38,39,70]. Of note, in comparison with other studies, microglia, after 1 h of incubation showed similar morphological characteristics to cells measured at peri-infarct cortical areas in experimental stroke models[50,71–73], using similar automated morphology analysis tools. Moreover, the ~50% drop in the number of microglial process endings observed in a study comparing control and Alzheimer's disease patients[74], closely resembles the changes that we observed in acute slices as early as after 20 min of incubation.

Importantly, microglial phenotype changes were found to represent a generic response to slice preparation as (i) comparable

morphological changes were seen in younger (P35 day) and older (P95 day) mice over time, or even during postnatal development (P18–20), (ii) different slice preparation techniques carried out in independent laboratories consistently resulted in a gradually increasing density of reactive microglia both at slice surfaces and in deeper layers. It has been previously suggested that gradually decreasing levels of oxygenation along the depth of the slice, as well as higher neuronal activity at the top surface could mediate injury-related signals, which may affect microglia distribution[75–77]. To visualize extracellular ATP in acute slices, we used a newly developed mouse line, where glutamatergic neurons express a highly sensitive fluorescent ATP-sensor in the plasma membrane. While it is well documented that microglial responses are markedly influenced by extracellular ATP and its derivatives through microglial NTPDase and P2Y12R mediated actions[37,45,70,78,79] and acute damage to the brain tissue releases large quantities of ATP[80–83], rapid microglial process recruitment to spontaneously emerging focal ATP events in acute slices was striking. While our results show elevated ATP-levels close to the cut slice surfaces as expected, they also indicate the formation of ATP gradients likely to drive microglial dislocation and morphological changes throughout the slices, including deeper tissue locations. While our observations reinforce general guidelines in electrophysiological investigations to avoid measurements taking place at the top 30–50 µm of slice preparations[84], we also demonstrate a unique form of a dynamic and sustained purinergic activity, featuring 1–2 min long flashing ATP release events in the nanomolar to micromolar range, deep in the slices even hours after slice cutting and similarly occurring in slices from both young (P18–20 days) or adult (P67–119 days) mice. The spatio-temporal characteristics of such ATP events are likely to reflect injury related astroglial activity, observed also in vivo[69,85]. However, due to the lack of appropriate tools to selectively block astrocytic ATP release in the acute slice model, we have not studied the source of ATP in the present paper. The decline in extracellular ATP levels is known to be mediated by NTPDase and Ecto-5′-nucleotidase enzymes, degrading ATP to ADP (the major P2Y12R ligand), AMP and adenosine, respectively[70,86]. Indeed, upon ecto-ATPase blockade we observed a gradual elevation of ATP levels that—presumably via ATP-induced ATP release—seemed to maintain, or even further enhance flashing ATP release events. It is also important to note that we observed regional differences in the ATP flash incidences and intensities, which could be due to either source cell or sensor (neuron) heterogeneity but may also reflect different responsiveness to injury at distinct brain locations, which remains to be investigated in future studies.

Microglia have critically important roles during development that can differ substantially from those of adult microglia[21]. We have also described that the early developmental formation of somatic purinergic junctions represents an important interface for microglia to monitor the status of immature neurons and control neurodevelopment[87]. Because the acute slice model is widely used in

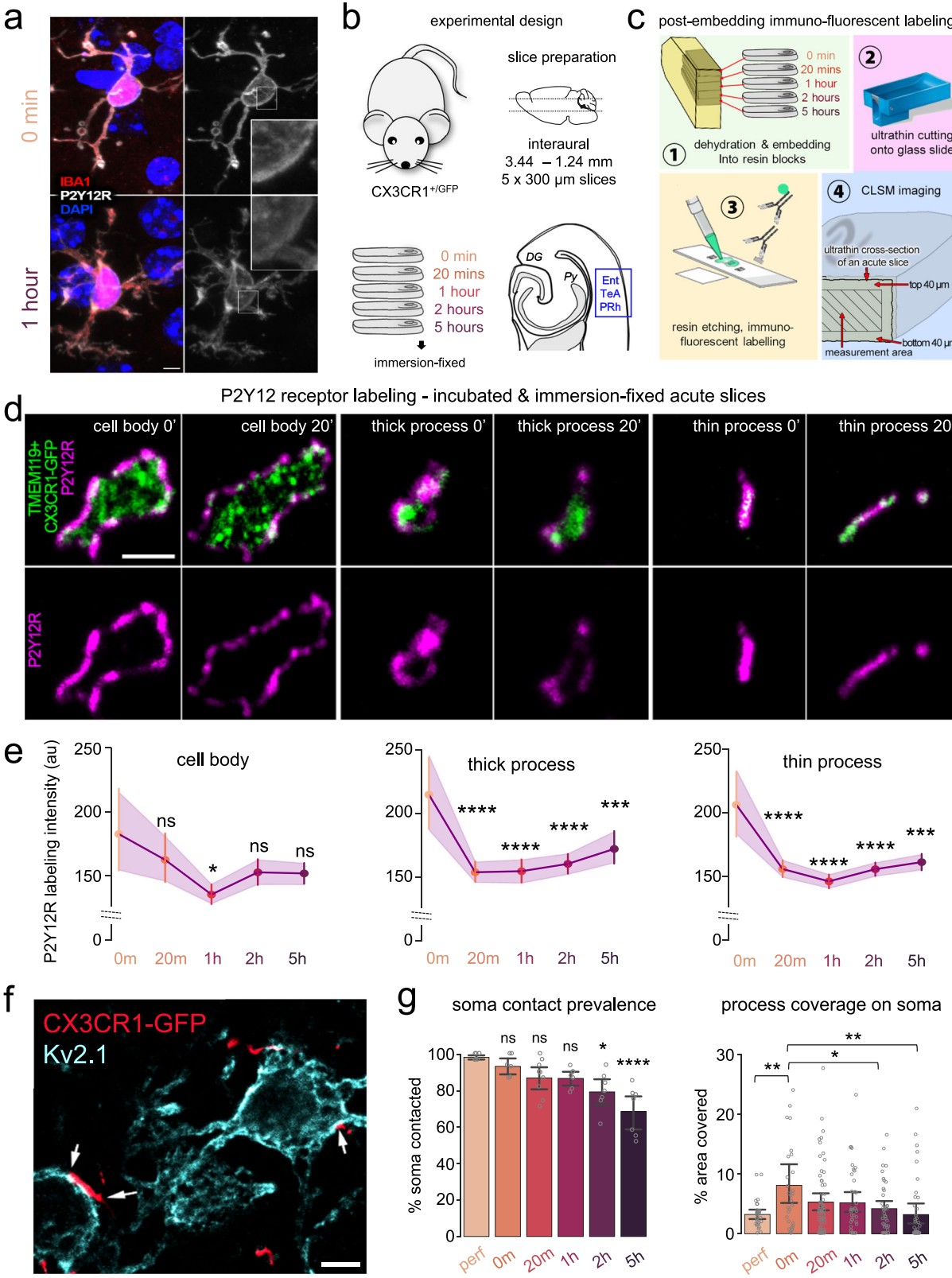

developmental brain research, we also performed a set of experiments on slices from postnatal mice. Interestingly, we found that microglia during late postnatal development behave in a very similar manner as in adulthood. The immediate ATP release caused by slice cutting or a later direct, secondary injury, the subsequent decay of the ATP levels at the injury site and the upcoming flashing ATP events were all observed on a similar time- and intensity-scale in slices from a wide (P19–P119)

age range. These data underscore the broad impact of the time-dependent changes we observed in this experimental model.

Microglia responded readily to changes in extracellular ATP (ADP) concentrations by P2Y12R-dependent fast morphological transformation and process redistribution. P2Y12 receptors not only represent a core marker for microglia in the healthy brain[88–90], but downregulation of P2Y12R is associated with reactive phenotypes and microglial

**Fig. 6 | Microglial P2Y12R undergoes rapid downregulation during the incubation process, paralleled by changes in microglia–neuron interactions.**
**a** CLSM z-stacks depicting microglial cells with pre-embedding immunofluorescent labelling (IBA1, red; P2Y12 receptors, white) before (0 min) and after 1 h of incubation. Scale bar: 3 μm (One independent experiment, $N = 1$ animal, $n = 1$ slice/animal, representative experiment). **b** CX3CR1$^{+/GFP}$ littermates ($N = 3$ animals, only males, P35 days) were used to create slice preparations, immersion-fixed at different time-points. Measurements were performed in Ent/TeA/PRh areas. **c** After fixation, slices were dehydrated and embedded into resin blocks (1), ultrathin slices were cut onto glass slides (2) and resin etching was followed by post-embedding P2Y12R immunofluorescent labelling (3). Finally, z-stack images were gathered from preparations via high-resolution CLSM (4). Each ultrathin section represents the whole cross-section of one acute slice (4), measurement can be done throughout the whole depth range of acute slices within a single image plane. **d** P2Y12R labelling via the post-embedding technique. Microglial cell bodies (left), thick processes (middle), thin processes (right) and P2Y12R labelling (top) or P2Y12R labelling only (bottom). Scale: 2 μm (one independent experiment from

$N = 3$ animal, $n = 3$ slices/animal). **e** Quantification of P2Y12R labelling intensity (arbitrary unit, see the "Methods" section) in the cell body (left), thick processes (middle) and thin processes (right.) $N = 3$ animal, P35 days; $n = 3$ slices/animal; One-way ANOVA, Dunnet's multiple comparisons, Body: $F(4, 40) = 2.746$, $p = 0.04$ (left), Thick: $F(4, 62) = 9.698$, $p < 0.0001$ (middle), Thin: $F(4, 154) = 13.09$, $p < 0.0001$ (right), mean ± SEM. Comparisons are made to 0 min values. Source data are provided as a Source Data file. **f** Representative section of MIP image created from z-stacks used to quantify microglial contact prevalence/process coverage on neuronal soma. White arrows: areas where microglial processes are likely to form contacts on neuronal soma (overlap of microglia and Kv2.1 labelling, Scale: 5 μm). CX3CR1$^{+/GFP}$ littermates (only males, P65 days). **g** Quantification of contact prevalence (left) and coverage (right) of neuronal soma by microglial processes ($N = 3$ animals, $n = 7$–9 regions of interest analysed/timepoint, 30-60 neurons/timepoint P65 days) One-way ANOVA, Dunn's multiple comparisons, $F(5,36) = 9.118$, $p < 0.0001$ (left), $F(5,235) = 3.409$, $p = 0.0054$ (right), mean ± SEM. Source data are provided as a Source Data file.

dysfunction both in vivo and ex vivo[37,53,56,91,92]. We show with a fully quantitative post-embedding method[59] that rapid downregulation of P2Y12R occurs most extensively at the processes of microglia already after 1 h of incubation proportionally with marked morphological changes. Importantly, this rapid and sustained decrease does not abolish the responsivity of microglia to flashing ATP events in the time window investigated (0–5 h), while reduced P2Y12R density was associated with altered microglia-neuron interactions. Purinergic signalling is thought to be involved in the regulation of overall slice activities, too, due to extracellularly produced or glia-derived adenosine[14,93], which contributes to an A1 receptor-mediated suppression of neuronal activity[15,94], a mechanism that could explain quiet states during recovery periods and baseline onset in acute slices and might contribute to inter-laboratory or inter-slice variability. Also, it is important to note, that glial ATP/adenosine release is required for closing/reopening windows of plasticity during development, both in the hippocampus[95] and the somatosensory cortex[96,97]. Important questions for future research will be how these local, dynamic changes in extracellular ATP, and consequently, adenosine levels could influence microglial and neuronal physiology in ex vivo slice preparations and whether such events are sufficient to re-initiate plasticity in the neuronal network in different forms of neuropathologies, in vivo.

In addition to the purinergic landscape, fractalkine–CX3CR1 signalling is another major microglia-related pathway involved in the regulation of microglia process redistribution in acute slices, driving microglial cell body translocation independently from P2Y12R. CX3CR1 signalling also appeared to differently regulate morphological changes of microglia compared to P2Y12R, delaying the transition to reactive phenotypes. Blockade of CX3CR1 receptors halts microglia process recruitment towards ATP flashes that may suggest a co-operation of the two pathways, maybe via the involvement of astroglial fractalkine production upon injury-related signals[98] in addition to neuronal fractalkine actions on microglia. Depending on the experimental model used, CX3CR1$^{+/-}$ mice in previous studies showed largely similar phenotypes to WT mice or intermediate phenotypes between WT and KO[99]. To consider this issue in our study, we have used either CX3CR1$^{+/-}$ mice as a microglia-reporter line—in these cases, the experiments were also repeated on CX3CR1 sufficient mice - or we used acute pharmacological inhibition of CX3CR1; while in experiments aimed to investigate the involvement of fractalkine signalling we compared CX3CR1-KO to wild-type mice.

The resting membrane potential of microglia has also been shown to correlate with morphology, as modified activity of tonically active K + channels responsible for maintaining resting membrane potential (knocking out THIK-1 or locally increasing extracellular [K+]) can decrease ramification[100,101]. It has been proposed that transitioning into a more reactive phenotype could partly reflect decreased expression

of THIK-1[101], as activation via LPS treatment resulted in a significantly downregulated expression of THIK-1 mRNA[102]. Importantly, our results indicate that gradual depolarization, rapid P2Y12R downregulation and transition of microglia into a reactive morphology are inherent features of acute slices, which have important implications for models using such preparations.

Changes in microglial distribution, morphology, P2Y12R expression, and local ATP fluxes are expected to markedly influence cell–cell interactions, given that P2Y12R signalling is required for normal microglial interactions with neuronal somata and synapses under different conditions[49,53,91,103]. We showed that while the prevalence of contacts on neuronal soma (somatic junctions) slowly and gradually decreases, the percentage of somatic area covered by microglial processes undergoes a more than two-fold increase immediately after slice preparation, similar to that observed in vivo after acute ischaemia[53]. These changes paralleled increased numbers of contacts on glutamatergic synapses, while contacts on GABAergic synapses were gradually decreasing. Studies have previously demonstrated that acute slice preparation induces synaptic sprouting[60–62], but the underlying mechanisms have not been revealed. Microglia are well-known regulators of synaptic density in vivo[104–108]. Of note, we found that the absence of microglia profoundly influenced the course of synaptic sprouting. While microglial contacts can facilitate spine formation of functional synapses both during development and in adult stages[105,107–109] the extent and speed of microglia-mediated effects in ex vivo brain slices were surprising. Of note, there are multiple signalling pathways implicated in microglia-synapse interactions (C1q, CD47, MHC1, etc.), the molecular elements of which are still not properly localized. Thus, further studies will be required to reveal the molecular mechanisms underlying these effects, potentially making the acute slice maturation process a valuable model for studying microglia-synapse interactions. The elevated GABAergic synaptic density at 0 min timepoint in the microglia-depleted mice could be explained by effects not related to slice cutting, as the lack of microglia has been already shown to elevate GABAergic synapse density in vivo[110].

Further studies would be important to explore sex differences that could potentially influence microglia-related effects observed in this study[111], in line with examining the same mechanisms at earlier stages of development (E15-P8), where microglial actions have been shown to be of critical importance[21]. Furthermore, the influence of different anaesthetics could also be tested, as more of these, such as the most widely used isoflurane has been shown to affect microglial behaviour and process dynamics[101]. Taken the observed robust microglial phenotype changes upon slice preparation and the fact that P2Y12R blockade almost fully prevented microglial process chemotaxis to focal ATP flashes suggest that isoflurane anaesthesia might

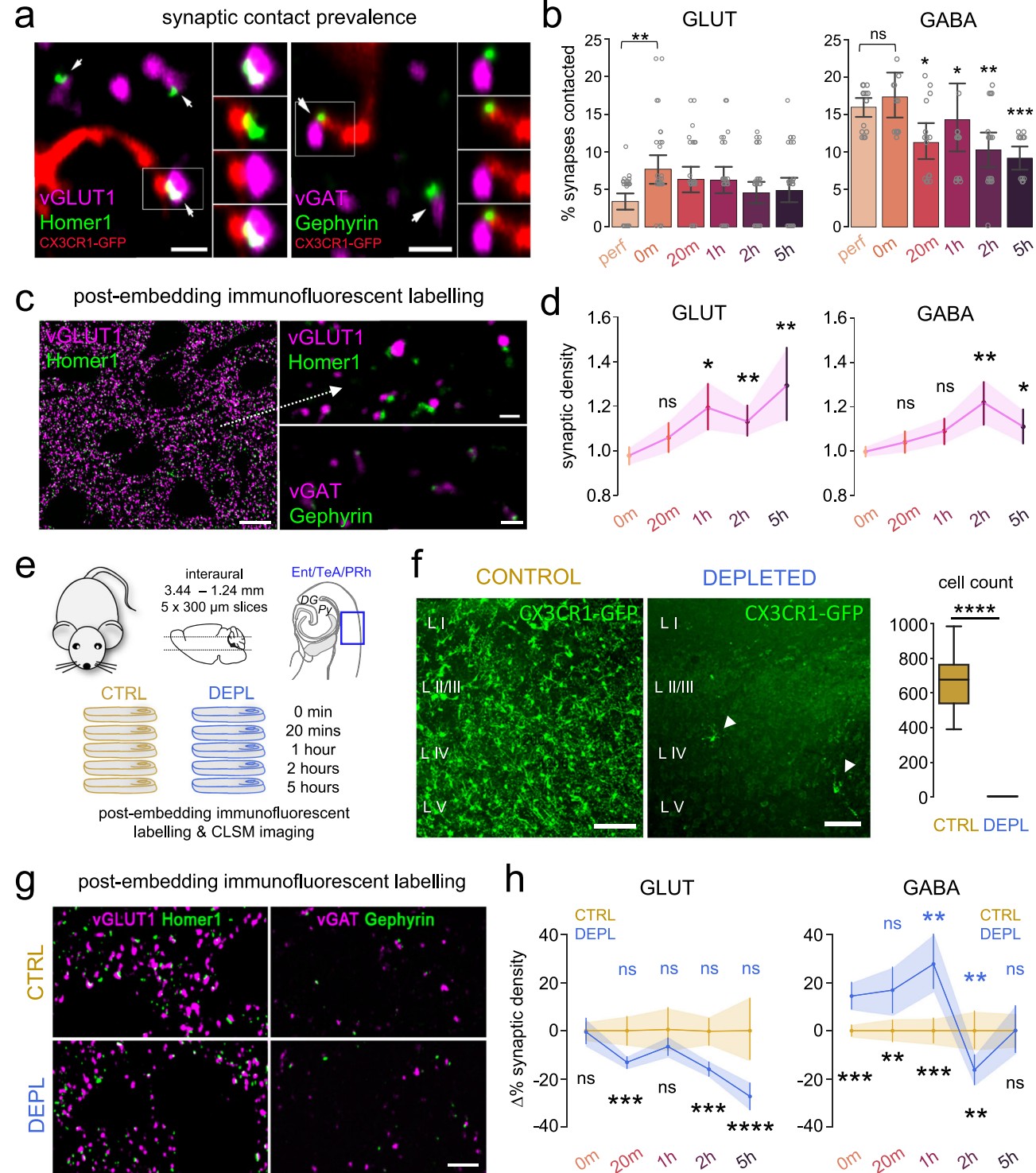

have had limited carryover effect to influence microglial behaviour in our experiments.

We show that the presence of functional microglia can actively influence SWR activity in hippocampal acute slice preparations, which was largely mediated by purinergic mechanisms. Thus, the presence of microglia and functional P2Y12R signalling are required for the emergence of physiological-like network activity in ex vivo slices suggesting that interventions aiming to inhibit microglial actions could render acute brain slices less capable of producing normal patterns of neuronal network activity seen in vivo. Previously it has been demonstrated that hippocampal field excitatory postsynaptic potentials

(fEPSPs) recover in correlation with synaptic changes in ex vivo slices[61]. Our results show that SWR activity shows a huge increase in amplitude, rate and frequency when correlated to the measured synaptic density changes. Since the generation of SWRs in these conditions are both reliant on a sufficiently large tonic excitatory activity and the action of reciprocally connected parvalbumin-positive basket cells[47], these results indicate that a delicate balance needs to be maintained between excitation and inhibition during the course of sprouting, which allows SWRs to emerge. Importantly, our results showed that microglia can facilitate glutamatergic and repress GABAergic synaptic formation to maintain a balanced reorganization of the network after

**Fig. 7 | Rapid changes in microglia–synapse interactions and microglia-dependent synaptic sprouting characterize acute slices. a** Single image planes from confocal z-stacks used to quantify contact prevalence of microglial processes on glutamatergic(left)/GABAergic(right) synapses. (Insets show channel pairs from boxes, pre- and postsynaptic marker, microglia and postsynaptic marker, microglia, and presynaptic marker, merged.) Arrows: individual synapses contacted by processes (bars: 1 μm). CX3CR1[+/GFP] littermates were used (only males, P65 days). **b** Microglial contact prevalence onto glutamatergic(left)/GABAergic(right) synapses ($N = 3$ animals, $n = 15-18$ ROI analysed/timepoint, P65 days) Kruskal–Wallis, Dunn's multiple comparisons, $p = 0.019$ (left), $p < 0.0001$ (right), mean ± SEM (compared to 0 min). Source data are provided as a Source Data file. **c** MIP images of confocal z-stacks used to quantify glutamatergic/GABAergic synaptic densities, created with post-embedding technique. Glutamatergic (left, bar: 5 μm) zoomed-in insets (right, top; bar: 1 μm), GABAergic (right, bottom; bar: 1 μm) labelling. **d** Synaptic density changes of glutamatergic (left)/GABAergic (right) synapses during incubation ($N = 6$ animal, $n = 6$ regions analysed/slice/timepoint, P65 days); Kruskal–Wallis, Dunn's multiple comparisons, $p = 0.0024$, mean ± SEM (compared

to 0 min). Source data are provided as a Source Data file. **e** CX3CR1[+/GFP] littermates (only males) were used to create control (CTRL; $N = 3$, P65 days; brown) and microglia depleted (DEPL; $N = 3$, P65 days; blue) subgroups. Slice preparations were obtained from both groups and immersion-fixed at different timepoints. **f** MIP images from control(left) or depleted(middle) slices (scale: 100 μm). Quantification comparing # of microglial cells in control(brown)/depleted(blue) slices ($N = 6$ animal/condition, P65 days); Mann–Whitney (two-sided), $p < 0.0001$, median ± SD. Boxes: interquartile range, whiskers: min–max, vertical bar: median. Source data are provided as a Source Data file. **g** MIP images created from confocal z-stacks used to compare densities of glutamatergic (left)/GABAergic (right) synapses in control (top) or depleted (bottom) slices (bar: 5 μm). **h** Comparison of glutamatergic(left)/GABAergic(right) density changes during incubation. (averages of control (brown, data also shown in **d**) vs. depleted condition (blue) ($N = 6$ animal/condition, $n = 6$ regions analysed/slice/timepoint, P65 days); Two-way RM ANOVA, Tukey's multiple comparisons, $F(4,184) = 4.668$, $p = 0.0013$ (left), $F(4,184) = 19.08$, $p < 0.0001$ (right), mean ± SEM. Source data are provided as a Source Data file.

the loss of synapses due to the slice preparation procedure. Supporting our data, partial depletion of microglia promoted asynchronous activity without an overall change in frequency[104], while microglia depletion in the hippocampus decreased spontaneously and evoked glutamatergic activity[112]. In contrast, in vivo imaging experiments showed increased excitatory and inhibitory neuronal activity in the cortex after microglia depletion, which results are in line with ex vivo circuit mapping data[113–115]. Thus, it is likely that in addition to the synaptic density changes, the altered distribution, morphology, and subsequent changes in microglia-neuron interactions are also key players in maintaining the integrity of neuronal networks in ex vivo acute slices, enabling complex, in vivo-like network activity patterns to emerge.

Interestingly, our in vivo chronic recordings confirm that microglial functions are important for the emergence of ripple activity not only in acute slices but also in the behaving animals. These data corroborate our ex vivo results, confirming that microglia are inherent modulators of complex neuronal networks, and their specific actions are indispensable to maintain neuronal network integrity and activity while also necessitating further research in order to better understand how microglial actions are specifically involved in complex neuronal network activity.

Acute slice preparation is a well-established experimental approach that is useful in studying the physiology of neurons and other cells, from individual synapses to complex neuronal networks. The fact that microglia in acute slices can present a rapid transition into a more reactive phenotype and can actively influence the neuronal network via altered microglia–neuron interactions, purinergic- and CX3CR1-mediated actions needs to be considered in ex vivo studies. It is important to note here that many other signalling pathways could influence microglia–neuron communication, including presently unknown forms of microglia–astrocyte interactions. Since microglial morphology strongly reflects the state of the tissue, monitoring these changes and comparison of microglial phenotype states in different slice preparation methods could facilitate more refined and consistent experimental models and paradigms to be established. Our results also highlight the importance of interactions between microglia and complex neuronal networks, which may be further emphasized by the sensitivity of microglia to a broad range of changes in their micro- and macroenvironment. While the observed microglial phenotypes in acute slices may not represent an undisturbed physiological state, the acute slice model also emerges as an instrumental tool to test and understand the different factors, that contribute to reactive microglial phenotypes or to assess how microglia-mediated changes impact complex neuronal networks, with broad implications for diseases of the CNS that are influenced by alterations of microglial function.

## Methods
### Animals
In our experiments, CX3CR1[+/GFP], CX3CR1[−/−], P2Y12[−/−], or C57Bl/6J mice littermates were used, except those in 2-photon imaging studies. For extracellular expression of the ATP sensor GRAB$_{ATP}$ in neurons, Vglut1/cre/Gt(Rosa26)Sor[cre/cre] mice[51] were crossed with LSL_GRAB$_{ATP}$_P2A_jRGECO1a(TgTm)−C57Bl/6J[flox/flox] mice made by Biocytogen (China) to generate the mouse strain ATP1.0$^{\Delta Vglut1}$. CX3CR1-cre/tdTomato mice were generated earlier in our laboratory[113]. All mice used in this study were male, with the exception of the data presented in Fig. 2c, Supplementary Fig. 1b, d, Supplementary Fig. 2g, h, where we used both genders in order to obtain sufficient animal numbers from littermates. Mice were kept in the vivarium on a 12-hour light/dark cycle and provided with food and water ad libitum. Temperature was $22 \pm 2\,°C$ and $50 \pm 10\%$ humidity. The animals were housed in two or three per cage.

### Slice preparation, incubation, and fixation
In order to minimize bacterial contamination, all the tools and containers used for slice preparation and incubation were routinely cleaned before and after experiments with 70% ethanol and were rinsed extensively with distilled water. For acute slice preparation, mice were decapitated under deep isoflurane anaesthesia. The brain was removed and placed into an ice-cold cutting solution, which had been bubbled with 95% $O_2$–5% $CO_2$ (carbogen gas) for at least 30 min before use. The cutting solution contained the following (in mM): 205 sucrose, 2.5 KCl, 26 NaHCO3, 0.5 CaCl2, 5 MgCl$_2$, 1.25 NaH$_2$PO$_4$, 10 glucose, saturated with 95% $O_2$–5% $CO_2$. Horizontal hippocampal slices of 300 μm or 450 μm (in case of LFP recordings) thickness were cut using a Vibratome (Leica VT1000S). The process of slice preparation from termination till the first slice to be immersion-fixed took ~5–10 min.

After acute slice preparation, slices were placed into an interface-type holding chamber for recovery. In an interface-type chamber, slices are laid onto a mesh just slightly submerged in the artificial cerebrospinal fluid (ACSF). Therefore the oxygenation of the tissue is mainly realized by the direct exposure to humidified oxygen-rich air above the slices. This chamber contained standard ACSF at 35 °C that gradually cooled down to room temperature. The ACSF solution contained the following (in mM): 126 NaCl, 2.5 KCl, 26 NaHCO$_3$, 2 CaCl$_2$, 2 MgCl$_2$, 1.25 NaH$_2$PO$_4$, 10 glucose, saturated with 95% $O_2$–5% $CO_2$. Immediately after slice preparation/given timeframes of incubation/after recordings, slices were immersion-fixed for 1 h with 4% PFA solution. In the case of PSB-treated acute slices, both cutting and standard ACSF solutions used to prepare the slices contained 10 μM PSB-0739 (Sigma-Aldrich).

For perfusion-fixed slices, mice were anaesthetised and transcardially perfused with 0.9% NaCl solution for 1 min, followed by 4% PFA in

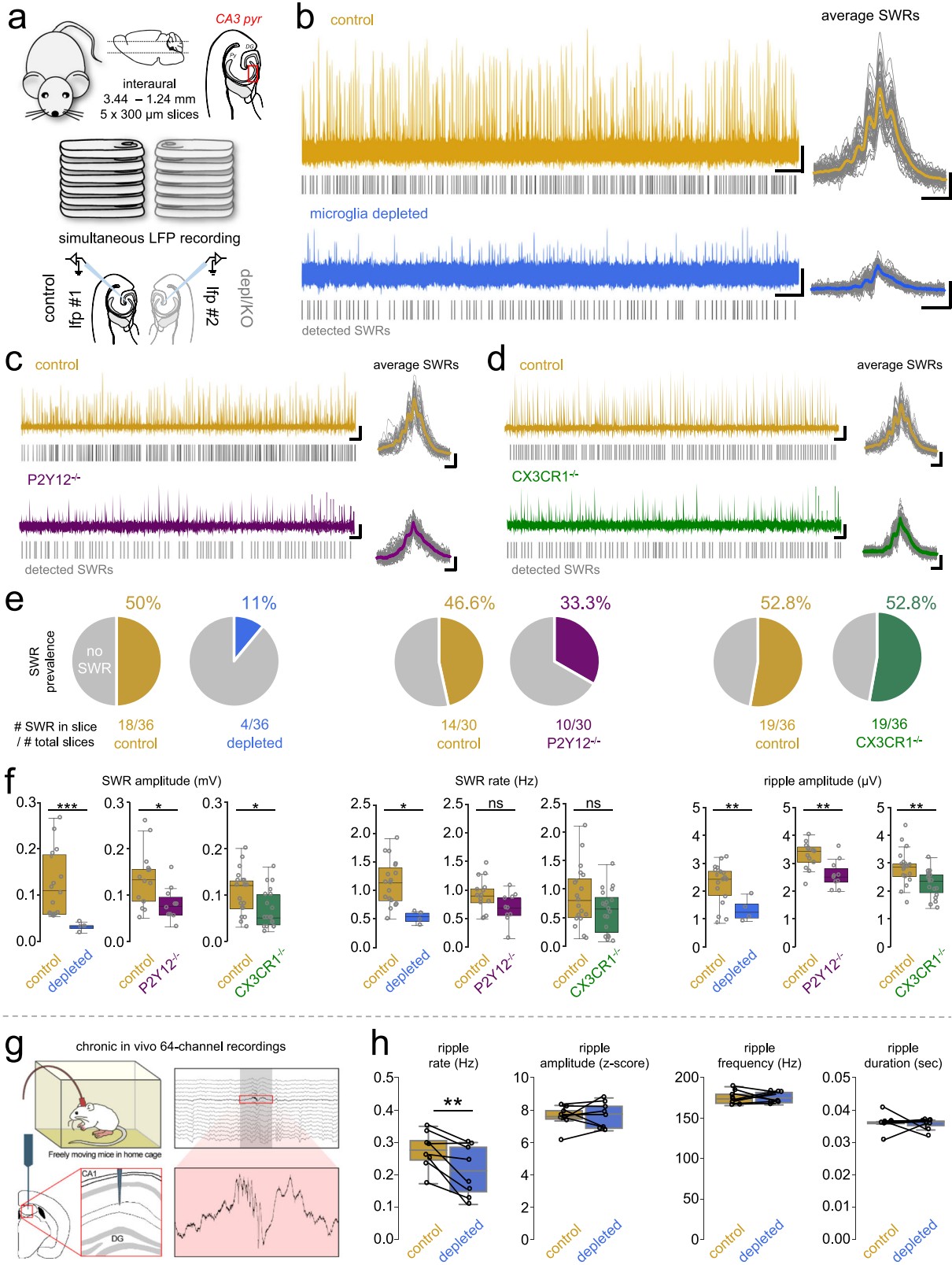

0.1 M phosphate buffer (PB) for 40 min, followed by 0.1 M PB for 10 min to wash the fixative out. Blocks containing the somatosensory cortex and ventral hippocampus were dissected, and horizontal sections were prepared on a vibratome (VT1200S, Leica, Germany) at 50 μm thickness for immunofluorescent histological and 100 μm thickness for the automated morphological analysis.

**Lab#2 slice preparation protocol**

Mice were anaesthetised with isoflurane and decapitated according to a protocol approved by the UCLA Chancellor's Animal Research Committee. After decapitation the head was immersed in ice-cold ACSF solution (see below) and put in the −80 °C freezer for 1 min. The brain was then removed from the skull and coronal 350 μm thick slices

**Fig. 8 | The absence of microglia, microglial P2Y12 or CX3CR1 dysregulates sharp wave-ripple activity. a** Acute hippocampal slices were obtained from control/experimental groups. Slices were measured in a pairwise manner, SWR activity was obtained from CA3 pyr. via LFP recordings. Depletion: CX3CR1$^{+/GFP}$ littermates (only males) were subjected to 3 weeks of either control/ PLX3397 diet (CTRL; $N = 6$, P65; gold, DEPL; $N = 6$, P65; blue) Knockouts: C57Bl/6J ($N = 5$ and $N = 6$) vs. P2Y12KO ($N = 5$) or CX3C1 KO ($N = 6$), P65 days). **b** Representative LFP recordings (left; bar: 30 s, 50 μV), average SWR events (right, $N = 50$, Scale: 100 ms, 25 μV) from control (top, gold) and depleted (bottom, blue) slices. Grey lines: individual SWR events. **c** Same as in (**b**), control (top, gold) and P2Y12R-KO (bottom, purple) slices. **d** Same as in (**b**), control (top, gold) and CX3CR1-KO (bottom, green) slices. **e** Pie-charts: SWR activity occurrence in control (gold), depleted (blue), P2Y12-KO (purple), and CX3CR1-KO (green) conditions. Grey: no spontaneous SWR activity ($N = 6$ ctrl vs. 6 depl animal, $N = 5$ ctrl vs. 5 P2Y12 KO animal, $N = 5$ ctrl vs. 5 CX3CR1 KO animal, 5 slices/animal, P65 days). **f** Quantification of SWR amplitude (left), rate (middle)

and Ripple amplitude (right) from control (gold), depleted (blue), P2Y12 KO (purple), and CX3CR1 KO (green) slices. ($N = 6$ ctrl vs. 6 depl animal, $N = 5$ ctrl vs. 5 P2Y12 KO, $N = 5$ ctrl vs. 5 CX3CR1 KO animal, 5 slices/animal, P65 days); Mann–Whitney (two-sided), $p < 0.001$. (Each group had its own controls from littermates.) Boxes: interquartile range, whiskers: min-max, vertical bar: median. Source data are provided as a Source Data file. g. In vivo experiment. 64-channel probes were chronically implanted into the hippocampus of mice and recorded in their home cage. After the depletion of microglia (same mice), animals were re-measured. Example of a 1 s multiple channel recording from stratum oriens (top) and stratum lacunosum-molaculare (bottom). 200 ms signal from one channel corresponding to pyramidal-cell layer, showing an identified ripple-event. **h** Quantification of in vivo ripple parameters. Paired $t$-test (two-sided), $p < 0.01$ ($N = 8$ animals, P93–136 days). Boxes: range, whiskers: min–max, vertical bar: mean. Source data are provided as a Source Data file.

were cut from a range of AP (from bregma): − 0.5 to −3 mm on a Leica VT1000S vibratome in ice-cold N-Methyl-ᴅ-Glutamine (NMDG)-based HEPES-buffered solution, containing (in mM): 135 NMDG, 10 ᴅ-glucose, 4 MgCl$_2$, 0.5 CaCl$_2$, 1 KCl, 1.2 KH$_2$PO$_4$, 20 HEPES, 27 sucrose (bubbled with 100% O$_2$, pH 7.4, 290–300 mOsm/L). Then, slices were incubated at 34 °C in a reduced sodium + sucrose artificial CSF (ACSF), containing (in mM): NaCl 85, ᴅ-glucose 25, sucrose 55, KCl 2.5, NaH$_2$PO$_4$ 1.25, CaCl$_2$ 0.5, MgCl$_2$ 4, NaHCO$_3$ 26, pH 7.3-7.4 when bubbled with 95% O$_2$, 5% CO$_2$ before fixation. Each slice was placed with the same orientation into an incubation chamber, and the slice surface was identified as the side facing toward the top. The incubation period for a series of slices from the same animal was designed as 0 min (this slice was also placed into the incubation chamber for 3–5 s), 20 min, 1, 2 and 5 h. After a certain time of incubation, the corresponding slice was carefully transferred from the incubation chamber to a 24-well plate pre-filled with 4% freshly depolymerized paraformaldehyde in 0.1 M phosphate buffer (PFA) for fixation. Each slice spent exactly 1 h in fixative before transferring into the "wash" plate and washed extensively with 0.1 PBS by exchanging the PBS in the well 3 times first, then placing the washing plate on a shaker and washing in PBS for 3 × 10 min. Slices were then individually inserted into 15 ml plastic test tubes sandwiched between tissue paper to avoid shaking during transportation. The test tubes were placed in a Styrofoam container and shipped to the IEM.

### Lab#3 slice preparation protocol
C57Bl/6J mice at the age of 4 weeks were decapitated under deep isoflurane anaesthesia. The brain was quickly removed and immersed in ice-cold sucrose-containing artificial cerebrospinal fluid, saturated with 95% O$_2$–5% CO$_2$ (sucrose-containing ACSF; in mM: NaCl 60, sucrose 100, KCl 2.5, NaH$_2$PO$_4$ 1.25, NaHCO$_3$ 26, CaCl$_2$ 1, MgCl$_2$ 5, ᴅ-glucose 20; pH 7.4, 310 mOsmol). Coronal hippocampal slices (300 μm thick) were prepared with a vibratome (VT1200S, Leica). The slices were incubated at 35 °C in the same solution for 20 min. The 0-min slice was placed into the incubation chamber for a few seconds (3–5 s) before fixation with 4% PFA. Then, 20 min slices were fixed with 4% PFA before being transferred to a second chamber filled with ACSF saturated with 95% O$_2$–5% CO$_2$ at room temperature (ACSF; in mM: NaCl 125, KCl 3.5, NaH$_2$PO$_4$ 120, NaHCO$_3$ 26, CaCl$_2$ 2, MgCl$_2$ 5, ᴅ-glucose 15; pH 7.4, 310 mOsmol). After that, 1, 2, and 5 h slices were fixed with 4% PFA. Each slice spent 1 h for fixation, and then transferred into wash plates containing 0.1 M phosphate buffer (PB) and washed with PB 3 times for 10 min. All slices were stored and transported submerged under 0.1 M PB, supplemented with 0.05 % sodium azide.

### Cross-section of slice preparations and quantification of translocation
300 μm-thick acute slices were immersion fixed immediately after slicing (0 min) or after 20 min, 1, 2 or 5 hours spent in an interface-type incubation chamber (Fig. 1a). Fixed slices were washed in 0.1 M PB, flat

embedded in 2% agarose blocks, rotated 90°, and resliced on a vibratome (VT1200S, Leica, Germany) at 50 μm thickness (Fig. 1b). The sections were mounted on glass slides, and coverslipped with Aqua-Poly/Mount (Polysciences). Intrinsic (CX3CR1$^{+/GFP}$) immunofluorescence was analysed using a Nikon Eclipse Ti-E inverted microscope (Nikon Instruments Europe B.V., Amsterdam, The Netherlands), with a CFI Plan Apochromat VC ×20 DIC N2 objective (numerical aperture: 075) and an A1R laser confocal system. We used 488 nm excitation laser (CVI Melles Griot), and image stacks (resolution: 0.62 μm/px) were taken with NIS-Elements AR. Maximal intensity projections of stacks containing the whole section thickness were saved in tiff format, cell bodies were masked with Fiji "Analyse particles" plugin. Cell-body masks were used to count cells, and these masks were subtracted from the original tiff files to get images containing microglial processes only. As a validation for volume-related quantifications in acute slices, the average thickness for each preparation across the incubation procedure was measured (mean ± SD: 262 ± 7 μm), which showed no substantial differences between groups (Kruskal–Wallis test, $p > 0.05$). For quantification, a measuring grid was placed onto the entire thickness of cross-sections (Fig. 1c), which divided the thickness into seven equal zones. Cell body numbers were counted within these grids, and microglial process volume was assessed by measuring fluorescent integrated density within the grids with Fiji software. Cell-body translocation calculation (Fig. 1f) was performed for assessing microglial movement towards the top surface and far away from the middle Z-depth of the slices. The coordinates for cell bodies were registered in Fiji software, and the distance of the cell bodies from the bottom surface or the (middle Z-depth) was measured at 0 min and at 5 h (the number of measured cells was identical at the two time-points). The distances were sorted in growing rows for both timepoints, and the 0-min values subtracted from the 5-h values, thus we could calculate the minimal values cells had to travel in order to reach the final distribution pattern (at 5 h) starting from the 0 min distribution. For the process area coverage measurement (Fig. 1l) the images with cell bodies masked out were used. Images from slices fixed at 0 min and 5 h were binarized in Fiji, and the percentage of the covered area was measured.

### Time-lapse imaging
Acute brain slices (300 μm thick) were prepared from 80-day-old CX3CR1$^{+/GFP}$ mice as described above. z-stack images (1 μm step size) were acquired using a Nikon C2 laser scanning confocal microscope equipped with a ×20 CFI Plan Apo VC (NA = 0.75 WD = 1.00 mm FOV = 1290.4 mm) objective at 488 nm, under continuous perfusion with ACSF (3 ml/min perfusion rate). The image acquisition started ~1 h after slice cutting. Image stacks were taken every 20 min for 6 h. Video editing was performed using NIS Elements 5.00 and ImageJ 1.53f51.

The same confocal setup was used in combination with a ×40 objective (CFI Apochromat Lambda S ×40/1.25 WI NA = 1.25 WD =

0.18 mm FOV (full) = 840 μm) to image ATP dynamics after ecto-ATPase blockade by ARL 67156 (Tocris; 10uM); 0.5 frame/s. To follow ATP dynamics before or after secondary mechanical injury to the slices, we used a CoolLed pE-4000 illumination system together with a Hamamatsu ORCA-Flash 4.0 camera, at ×4 magnification (Plan Fluor objective, NA = 0.13, WD = 17.2 mm, FOV = 3215.36 μm). Secondary injury was inflicted manually by a sterile blade and imaging started as soon as possible, no more than 1–2 min after injury. For these purposes, the ATP sensor was either delivered by AAV injections to the brains of CX3CR1-cre/tdTomato mice, or we used slices prepared from ATP1.0$^{\Delta Vglut1}$ mice. The age of mice in these experiments varied between P19-20 and P69-99 as specified in figure legends.

### Automated morphological analysis of microglial cells

300 μm-thick acute slices were immersion fixed for 1 h immediately after slicing (0 min) or after 20 min, 1, 2 or 5 h spent in an interface-type incubation chamber (Fig. 2d, e). Fixed slices were washed in 0.1 M PB, flat embedded in 2% agarose blocks, and re-sectioned on a vibratome (VT1200S, Leica, Germany) at 100 μm thickness (Fig. 1b). Sections selected from the middle region of incubated slices were immunostained with antibodies, and DAPI (for primary and secondary antibodies used in this study, please see Supplementary Table 1). Preparations were kept in a free-floating state until imaging to minimize the deformation of tissue due to the mounting process. Imaging was carried out in 0.1 M PB, using a Nikon Eclipse Ti-E inverted microscope (Nikon Instruments Europe B.V., Amsterdam, The Netherlands), with a CFI Plan Apochromat VC ×60 water immersion objective (numerical aperture: 1.2) and an A1R laser confocal system. Volumes were recorded with 0.2 μm/pixel resolution and a Z-step of 0.4 μm. For 3-dimensional morphological analysis of microglial cells, the open-source MATLAB-based Microglia Morphology Quantification Tool was used (available at https://github.com/isdneuroimaging/mmqt). This method uses microglia and cell nuclei labelling to identify microglial cells. Briefly, 59 possible parameters describing microglial morphology are determined through the following automated steps: identification of microglia (nucleus, soma, branches) and background, creation of 3D skeletons, watershed segmentation and segregation of individual cells[50].

### Pre-embedding immunofluorescent labelling and analysis of CLSM data

Before the immunofluorescent labelling, the 50 μm-thick sections were washed in PB and Tris-buffered saline (TBS). Thorough washing was followed by blocking for 1 h in 1% human serum albumin (HSA; Sigma-Aldrich) and 0.03–0.1% Triton X-100 dissolved in TBS. After this, sections were incubated in mixtures of primary antibodies and diluted in TBS overnight at room temperature. After incubation, sections were washed in TBS and incubated overnight at 4 °C in the mixture of secondary antibodies, all diluted in TBS. Secondary antibody incubation was followed by washes in TBS and PB; the sections were mounted on glass slides, and coverslipped with Aqua-Poly/Mount (Polysciences). Immunofluorescence was analysed using a Nikon Eclipse Ti-E inverted microscope (Nikon Instruments Europe B.V., Amsterdam, The Netherlands), with a CFI Plan Apochromat VC ×60 oil immersion objective (numerical aperture: 1.4) and an A1R laser confocal system. We used 405, 488, 561 and 647 nm lasers (CVI Melles Griot), and scanning was done in line serial mode, pixel size was 50 × 50 nm. Image stacks were taken with NIS-Elements AR. For primary and secondary antibodies used in this study, please see Supplementary Table 1. Quantitative analysis of each dataset was performed by at least two observers, who were blinded to the origin of the samples and the experiments and did not know of each other's results.

For the analysis of somatic contact prevalence, confocal stacks with double immunofluorescent labelling (cell type marker and microglia) were acquired from at least three different regions of the mouse cortex. All labelled and identified cells were counted when the whole cell body was located within the z-stack. Somata were considered to be contacted by microglia when a microglial process clearly touched it (i.e., there was no space between neuronal soma and microglial process) on at least 0.5 μm long segment.

Microglial process coverage was measured on CLSM z-stacks acquired with a step size of 300 nm. On single-channel images, Kv2.1-positive cells were selected randomly, the cell bodies of which were fully included in the captured volume. The surface of these cells was calculated by measuring the circumference of the soma on every section multiplied by section thickness. The surface of microglial process contacts was measured likewise.

For the analysis of synaptic contact prevalence, confocal stacks with triple immunofluorescent labelling (pre- and postsynaptic markers and microglia) were analysed using an unbiased, semi-automatic method. First, the two channels representing the pre- and postsynaptic markers were exported from a single image plane. The threshold for channels was set automatically in FIJI, the "fill in holes" and "erode" binary processes were applied. After automatic particle tracking, synapses were identified where presynaptic puncta touched postsynaptic ones. From these identified points we selected a subset in a systematic random manner. After this, the corresponding synapses were found again in the original z-stacks. A synapse was considered to be contacted by microglia when a microglial process was closer than 200 nm (4 pixels on the images).

### Post-embedding immunofluorescent labelling and quantitative analysis

The technique described by Holderith et al.[59] was used with slight modifications. 300 μm-thick acute slices were cut from CX3CR1$^{+/GFP}$ mouse line and then immersion fixed immediately after slicing (0 min) or after 20 min, 1, 2 or 5 h spent in an interface-type incubation chamber (Fig. 1b). Fixed slices were washed in 0.1 M PB and 0.1 M maleate buffer (MB, pH: 6.0). Then slices were treated with 1% uranyl-acetate diluted in 0.1 M MB for 40 min in dark. This was followed by several washes in 0.1 M PB, then slices were dehydrated in ascending alcohol series, acetonitrile and finally embedded in Durcupan (Fluca). Each block contained all slices from a respective time series of one animal (Fig. 6c/1). Ultrathin sections were cut using a Leica UC7 ultramicrotome at 200 nm thickness in such a way that each and every one of these ultrathin sections contained the full width of all acute brain slices and collected onto Superfrost Ultra plus slides and left on a hotplate at 80 °C for 30 min then in the oven at 80 °C overnight (Fig. 6c/2). Sections were encircled with silicon polymer (Body Double standard kit, Smooth-On, Inc.) to keep incubating solutions on the slides. The resin was etched with saturated Na-ethanolate for 5 min at room temperature. Then sections were rinsed three times with absolute ethanol, followed by 70% ethanol and then DW. Retrieval of the proteins was carried out in 0.02 M Tris Base (pH = 9) containing 0.5% sodium dodecyl sulfate (SDS) at 80 °C for 80 min. After several washes in TBS containing 0.1% Triton X-100 (TBST, pH = 7.6), sections were blocked in TBST containing 6% BlottoA (Santa Cruz Biotechnology), 10% normal goat serum (NGS, Vector Laboratories) and 1% BSA (Sigma) for 1 h then incubated in the primary Abs diluted in blocking solution at room temperature overnight with gentle agitation (Fig. 6c/3). After several washes in TBST, the secondary Abs were applied in TBST containing 25% of the blocking solution for 3 h. After several washes in TBST, slides were rinsed in DW then sections were mounted in Slow-fade Diamond (Invitrogen) and coverslipped. Immunofluorescence was analysed using a Nikon Eclipse Ti-E inverted microscope (Nikon Instruments Europe B.V., Amsterdam, The Netherlands), with a CFI Plan Apochromat VC ×60 oil immersion objective (numerical aperture: 1.4) and an A1R laser confocal system. We used 488 and 647 nm lasers (CVI Melles Griot), and scanning was done in line serial mode, pixel size was 50 × 50 nm. Image stacks were taken with NIS-Elements AR. For

primary and secondary antibodies used in this study, please see Supplementary Table 1. Quantitative analysis of each dataset was performed by at least two observers, who were blinded to the origin of the samples and the experiments and did not know of each other's results.

For the quantitative assessment of P2Y12R expression, single high-resolution confocal laser-scanning microscopy (CLSM) image planes were used. As each ultrathin section represents the whole cross-section of one acute slice (Fig. 6c/4), measurement could be done throughout the whole depth range of acute slices within a single image plane. We performed the measurements in the middle parts of the slices, avoiding the top and bottom 40 μm. Microglial cell bodies, thick (average diameter >1 μm) and thin (average diameter <1 μm) processes were identified based on TMEM119 and CX3CR1$^{+/GFP}$ staining. Once the respective outlines of these profiles have been delineated, these outlines have been extended both in the intra- and the extracellular direction with 250–250 nm, yielding a 500 nm wide ribbon-shaped ROI. The integrated fluorescent density of P2Y12R-labelling was measured and divided by the lengths of the respective ROIs, which gave us the P2Y12R fluorescent intensity values applied to unit membrane lengths for each profile.

For the synapse density measurements, we used double stainings for pre- and postsynaptic markers (vGluT1 with Homer1 for glutamatergic, and vGAT with Gephyrin for GABAergic synapses). We could validate the specificity and sensitivity of these stainings based on the near-perfect match between pre and postsynaptic markers, thus we continued to measure the presynaptic signals. ROIs were randomly chosen within the neuropil, avoiding cell bodies. The integrated fluorescent densities were measured within these ROIs for vGluT1 and vGAT channels. Measurements were performed with the Fiji software package.

### Ex vivo 2-photon imaging of extracellular ATP

ATP1.0$^{\Delta Vglut1}$ mice were used in the ex vivo 2-photon imaging experiments at the age of P19–20 or P67–74 days. Acute slices were obtained as in previous experiments (see the "Methods" subsection "Slice preparation, incubation, and fixation"). Slices, including the hippocampus and cortical regions mentioned above, were used for two-photon imaging (5 slices/brain). The slices were split, and from each plane, one half was used for imaging within the CA3 area in the hippocampus and the other half for imaging within layer two in the cortex, both in a 256*128 μm window. The time between slice cutting and imaging was minimized to 8–13 min for the very first slices. After that, new slices were imaged each hour and kept in the carbogenated incubation chamber at room temperature (~20 °C) until use. For each slice, 10 min imaging at 0.5 frame/s rate was followed by z-stack acquisition (3 μm step size; taking ~3–4 min, see experimental setup in Fig. 4a). Two-photon imaging was accomplished by a Nikon Eclipse FN1 upright microscope, equipped with a Nikon A1R MP+ multiphoton system using a Chameleon Vision II Ti: Sapphire tuneable (680–1080 nm) laser and a ×25 water dipping objective lens [CFI75 Apo LWD NA = 1.1 WD = 2 mm]. The excitation wavelength was set to 920 nm for detecting the signal, and a 550/88 nm emission filter and a GaAsP NDD PMT detector were used. Time-lapse data were analysed using the NIS-Elements (version 5.42.02) and ImageJ (version 1.53t) software. Mean Fluorescent Intensity (MFI) decreases (Fig. 4b, c) were measured in areas where no ATP flash activity (focal ATP surges) was observed [ROI size: 12.5*12.5 μm; 4 ROI/slice]. z-stack images were used to determine the fluorescent signal decrease in time and depth (Fig. 4d, e, Supplementary Fig. 2c), by averaging MFI data from three regions [ROI size: 25*50 μm] from each slice. ATP flashes were identified on each slice manually, encircling the individual flashes when the signal intensity was at maximum. Then, flash size, mean fluorescent intensity, peak duration, rise time and decay time data were collected and analysed. Details of statistical analysis are provided in Fig. 4 legend and Supplementary Fig. 2.

For studying microglia behaviour in response to extracellular ATP events we injected CX3CR1-cre/tdTomato mice (P96-119, generated earlier[113]) bilaterally with 3 × 100 nL of AAV9-hsyn-cATP1.0 (chick) construct (WZ Biosciences; 1*10$^{13}$vg/ml) at the lateral cortex (coordinates in respect to bregma: AP: 3, ML: 3,6, DV: 4, 3.2, 2.5 mm, 100 nL/depth). Mice were sacrificed 5–7 weeks after virus injection, and acute slices were used for 2P imaging similarly to that described above (256 × 128 μm image size, 0.5 frame/s rate, three Z steps 4 μm apart), with the exception that the excitation wavelength was set to 930 nm. Then, individual ATP events were extracted from maximal intensity projection time-lapse images and ATP flash properties and microglia displacement were measured as indicated in Fig. 5e. In brief, intensity/area/duration/latency of ATP events were extracted from z-stack acquisitions via the NIS–Elements software and calculated via custom written software in python (environment 2.7.0 by D.S.) Changes in the tdTomato signal corresponding to directed microglial processes were calculated as the % change in baseline (average of the signal 100 s before the ATP events) compared to the signal value at the closest peak detected to the peak in the cATPs signal (directed movement peak max) or when there were no definitive detectable peaks 300 s after the peak in cATPs signal, 100 s of the tdTomato signal was averaged after the cATPs signal peak. In some experiments, slices were pre-incubated with 4 μM PSB0739 (Tocris) or 10 μM AZD8797 (MedChemExpress) for 5 min. To calibrate the ATP sensor, we used standard recording pipettes (see: patch-clamp recordings) filled with 10, 50, 100 nM or 0.5, 1, 5 μM ATP diluted in standard ACSF (the same solution which we used in the recording chamber) and positioned them in the tissue, while delivering gentle puffs during recording. k-means clustering of the data in Figs. 4 and 5 were calculated by using the scikit learn: StandardScaler and KMeans packages[116].

### Selective depletion of microglia

CX3CR1$^{+/GFP}$ or C57Bl/6J littermates were subjected to 3 weeks of either control or PLX3397-containing diet to create a microglia-depleted subgroup of animals, respectively. Extra slices were gathered from each animal by the re-slicing of immersion-fixed or perfusion-fixed acute slice preparations (50 μm). Success of depletion was monitored by creating z-stack images of the native GFP signal of microglia with confocal laser-scanning microscopy and verified by comparing the total number of microglia counted (via Fiji counting tool) at the same cortical and hippocampal locations and through the whole depth of the slice preparations.

### Patch-clamp recordings

Generally accepted guidelines were followed for patching microglial cells[84]. After incubation for given timeframes (as specified for each experiment in the "Results" section), slices were transferred individually into a submerged-type recording chamber with a superfusion system allowing constantly bubbled (95% O$_2$–5% CO$_2$) ACSF to flow at a rate of 3–3.5 ml/min. The ACSF was adjusted to 300–305 mOsm and was constantly saturated with 95% O$_2$–5% CO$_2$ during measurements. All measurements were carried out at 33–34 °C, the temperature of the ACSF solution was maintained by a dual-flow heater (Supertech Instruments). The pipette solution contained (in mM): 120 KCl, 1 CaCl$_2$, 2 MgCl$_2$, 10 HEPES, and 11 EGTA, pH: 7.3, 280–300 mOsm. Pipette resistances were 3–6 MΩ when filled with pipette solution. Visualization of slices and selection of cells (guided by native GFP signal) was done under an upright microscope (BX61WI; Olympus, Tokyo, Japan equipped with infrared-differential interference contrast optics and a UV lamp). Only cells located deeper than -50 μm (measured from the slice surface) were targeted. All cells were initially in voltage-clamp mode and held at −40 mV holding potential during the formation of the gigaseal. Series resistance was constantly monitored after the whole-cell configuration was established, and individual recordings taken for analysis showed stability in series resistance between a 5%

margin during the whole recording. After the whole-cell configuration was established, resting membrane potential values were measured by changing the recording configuration to current-clamp mode at 0 pA for a short period of time (10–15 s) and evaluated from the recorded signal via averaging a 5 s period. Thereafter, responses to a pulse train of current steps (−2 to −10 pA with 2 pA increments and 10 ms duration) were recorded. Quantification of the input resistance of cells was derived via Ohm's law based on the slope of voltage responses measured at each current step. The inter-pulse interval was 100 ms. Recordings were performed with a Multiclamp 700B amplifier (Molecular Devices). Data were digitized at 10 kHz with a DAQ board (National Instruments, USB-6353) and recorded with a custom software developed in C#.NET and VB.NET in the laboratory. Analysis was done using custom software developed in Delphi and Python environments.

## LFP recordings

Acute slice preparations were gathered at each recording day (6 slices/animal, 450 μm thick) in a pairwise manner from control and microglia-depleted animals while using the same solutions and equipment. The slice preparation sequence was alternated throughout the recording days between the two groups, as well as the chambers that were used for the incubation process, to minimize artefacts that might have been introduced by variance in slice preparation or incubation quality. Each experimental group had its own controls from littermates. After at least 1 h of incubation, slices from both conditions were transferred together in a pairwise manner to a dual perfusion system recording chamber[43], and measured simultaneously via performing local field potential (LFP) recordings. In this design, the slices were placed on a metal mesh, and two separate fluid inlets allowed ACSF to flow both above and below the slices at a rate of 3–3.5 ml/min for each flow channel at 33–34 °C (Supertech Instruments). The position of slices from the two conditions was also alternated in the recording chamber between subsequent measurements. Standard patch pipettes filled with ACSF were used for LFP recordings. Recording pipettes were positioned at the same depth (-80–100 μm below the surface) and in the same region (pyramidal layer of CA3) in both conditions. ACSF containing pipette resistances were 3–6 MΩ. Recordings were performed with a Multiclamp 700B amplifier (Molecular Devices). Data were digitized at 10 kHz with a DAQ board (National Instruments, USB-6353) and recorded with software developed in C#.NET and VB.NET in the laboratory.

## Digital signal processing and analysis

All data were processed and analysed off-line using self-developed programmes written in Delphi 6.0 by A.I.G. and Python 2.7.0. by P.B. Signals were filtered with a two-way RC filter to reserve phase. SWRs were pre-detected on 30 Hz low-pass-filtered field recordings using a threshold value of 2–3 times the SD of the signal. Recordings were considered to not contain SWRs if 2 times the SD of the signal did not result in detectable events. All automatic detection steps were supervised in each recording. The predetected SWRs were then analysed using a programme that measured various SWR features and eliminates recording artefacts similar to SWRs. Namely, on the low-pass-filtered signal, the programme measured: peak amplitude of sharp waves (SWR amplitude); inter- sharp wave interval (SWR rate). On a ripple bandpass-filtered trace (200 ± 30 Hz), the programme also detected the time of negative ripple peaks. Based on this, we identified the ripple cycle closest to the SWR peak and used its negative peak as a triggering event for averages to preserve the ripple phase. Taking the absolute value of the ripple bandpassed signal and low pass filtering it, we calculated the ripple power peak (Ripple amplitude). After detection, -100 consecutive events were selected for quantification, where the highest values were measured along the whole recording.

## In vivo chronic recordings

To study microglia-neuron interactions at individual neurons, we have established intracranial LFP measurements in freely behaving mice using 64-channel silicon probes (A1x64-Poly2-6mm-23s-160-H64_30mm, NeuroNexus). 4–5 months old C57BL/6 mice underwent surgical implantation of polytrodes under isoflurane anaesthesia, hippocampal probes were mounted on custom-made, adjustable microdrives, and the probe-microdrive assemblies were shielded by copper mesh, preventing the recordings from contamination by environmental electric noise. The mesh was covered by dental acrylic. The craniectomy was sealed by Dura-Gel (Cambridge NeuroTech) to prevent the damage of tissue and probe. Before finishing the surgery, Buprenorphine (dose, 0.045 μg/g body weight) was injected subcutaneously. Recordings were started after a one-week-long post-surgery recovery and habituation to connectorization. Recording sites spanned from the primary somatosensory cortex to the str. radiatum of the dorsal hippocampal CA1 region. Mice were observed in their home cages, and signals were acquired using an RHD2000 interface board (Intan Technologies). The movement of the animals was tracked with six high-speed cameras (Flex13, Opti-Track), using an infrared marker-based approach. After control recordings, mice were fed a chow diet containing the CSF1 receptor antagonist, PLX5622 (Plexxikon Inc.) for 3 weeks to eliminate microglia from the brain, and recordings were repeated. Evaluation was carried out using custom-written MATLAB functions. Probe positions and depletion success were evaluated after transcardial perfusion of the mice.

## Quantification and statistical analysis

All quantitative assessments were performed in a blind manner. Sample size was determined based on sample size calculations performed in our previous experiments using similar models. Data were sampled in a systematic random manner. Experiments were replicated by using multiple animals for slice preparations or histology (biological replicates), and pooled results from experiments were presented in the figures. Exclusion criteria were pre-established for the quality of acute slices and immunostainings. No samples had been excluded in the present paper. Normality of the data was tested via the Shapiro–Wilk test to decide whether to use parametric or non-parametric tests during further analysis. In the case of two independent groups, Student's $t$-test, or Mann–Whitney $U$-test for three or more independent groups, one-way ANOVA, two-way ANOVA or repeated-measures ANOVA was applied with Dunn's, Dunnet's or Tukey's multiple comparison tests. Statistical analysis was conducted in GraphPad Prism version 10.0.0 for Windows, GraphPad Software, Boston, Massachusetts, USA, www.graphpad.com. In this study, data are presented as mean ± SEM or in median–Q1–Q3 format, $p < 0.05$ was considered statistically significant. Significance levels are shown as follows: ns: $p > 0.05$; *$p < 0.05$; **$p < 0.01$; ***$p < 0.001$; ****$p < 0.0001$.

## Ethics statement

All experiments were approved by the Ethical Committee for Animal Research at the HUN-REN Institute of Experimental Medicine, Hungarian Academy of Sciences, and conformed to Hungarian (1998/XXVIII Law on Animal Welfare) and European Communities Council Directive recommendations for the care and use of laboratory animals (2010/63/EU) (license number PE/EA/2552-6/2016; PE/EA/254-7/2019). Lab#2: All procedures were performed in accordance with protocols approved by the UCLA Institutional Animal Care and Use Committee (IACUC, Protocol# ARC-1995-045) and guidelines of the National Institutes of Health. Lab#3: Experiments in Bonn followed the institutional and federal guidelines on animal care and use. (An animal license number is not required for ex-vivo experiments with brain slices.)

## Reporting summary

Further information on research design is available in the Nature Portfolio Reporting Summary linked to this article.

## Data availability

All data reported in this paper will be shared by the lead contact upon request. Any additional information required to reanalyse the data reported in this paper is available from the lead contact upon request, and all the data is provided as a Source Data file with this paper. Source data are provided with this paper.

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

## Acknowledgements

We thank László Barna, Pál Vági and the Nikon Imaging Center at the HUN- REN Institute of Experimental Medicine (IEM) for kindly providing microscopy support. We also thank the Cell Biology Center and the Virus Technology Unit of the IEM for providing technical support and to Dr. Eszter Sipos for the injection of AAV constructs. We thank Zsolt Kohus for methodological concepts and preliminary electrophysiological data. We are also grateful to Norbert Hájos (IEM), Zoltán Nusser (IEM), Dániel Schlingloff (IEM) and János Szabadics (IEM) for their support and useful comments, and Viktor Varga (IEM) for initial discussions regarding in vivo measurements. This work was supported by "Momentum" research grant from the Hungarian Academy of Sciences (LP2016-4/2016, and LP2022-5/2022 to A.D.) ERC-CoG 724994 (to A.D.) and Hungarian Brain Research Programme NAP2022-I-1/2022 a (to A.D.) the Hungary–China Intergovernmental Science and Technology Innovative Cooperation project (2021YFE0116400 to M.J. and 2020-1.2.4-TÉT-IPARI-2021-00005 to A.D.) and Project no. RRF-2.3.1-21-2022-00011, National Laboratory of Translational Neuroscience, which has been implemented with the support provided by the Recovery and Resilience Facility of the European Union within the framework of Programme Széchenyi Plan Plus (to A.D.). C.C. was supported by the János Bolyai Research Scholarship of the Hungarian Academy of Sciences. C.C. (UNKP-22-5) and B.P. (UNKP-22-4-I) were supported by the New National Excellence Programme of the Ministry for Innovation and Technology. I.M. was supported by the National Institutes of Health/National Institute of Aging grant R01050474 and the Coelho Endowment. H.B. was supported by SFB 1089 and EXC 2151 under Germany's Excellence Strategy, both of the Deutsche Forschungsgemeinschaft (DFG, German Research Foundation).

## Author contributions

Experimental design and overall concept, A.D., C.C., A.I.G., P.B.; Methodology, C.C., P.B., B.P., Z.K., E.S., An.D., H.B., K.A., I.M., X.W., K.H., Y.W., Z.W., M.J., Y.L, N.L.; Formal Analysis C.C., P.B., B.P., Z.K., An.D.; Investigation, P.B., C.C., B.P., E.S., Z.K., A.K., M.N; Resources, A.D., A.I.G. Writing—original draft, P.B., C.C., A.D., Z.K.; Editing, P.B., C.C., B.P., A.D., Z.K., H.B., K.A. I.M., X.W.; Visualization P.B., C.C., B.P., Z.K. Supervision A.D. and A.I.G.; Project Administration A.D.; Funding Acquisition, A.D.

## Competing interests

The authors declare no competing interests.

## Additional information

[1]János Szentágothai Doctoral School of Neuroscience, Semmelweis University, Budapest H-1083, Hungary. [2]Laboratory of Cerebral Cortex Research, HUN-REN Institute of Experimental Medicine, Budapest H-1083, Hungary. [3]Laboratory of Neuronal Network and Behaviour, HUN-REN Institute of Experimental Medicine, Budapest H-1083, Hungary. [4]Momentum Laboratory of Neuroimmunology, HUN-REN Institute of Experimental Medicine, Budapest H-1083, Hungary. [5]Laboratory of Thalamus Research, HUN-REN Institute of Experimental Medicine, Budapest H-1083, Hungary. [6]Department of Neurology, The David Geffen School of Medicine at UCLA, Los Angeles, CA 90095, USA. [7]Institute of Experimental Epileptology and Cognition Research, Medical University of Bonn, Bonn 53127, Germany. [8]University Hospital Bonn, Bonn, Germany. [9]State Key Laboratory of Membrane Biology, New Cornerstone Science Laboratory, School of Life Sciences, Peking University, 100871 Beijing, China. [10]Chinese Institute for Brain Research, 102206 Beijing, China. [11]State Key Laboratory of Molecular Developmental Biology, Institute of Genetics and Developmental Biology, Chinese Academy of Sciences, 100101 Beijing, China. [12]These authors contributed equally: Péter Berki, Csaba Cserép, Zsuzsanna Környei. ✉e-mail: denes.adam@koki.hu

