## [Peer Review File · Nature Communications]

REVIEWER COMMENTS

Reviewer #1 (Remarks to the Author):

The authors present an improved version of the manuscript formerly entitled “Microglia undergo rapid phenotypic transformation in acute brain slices but remain essential for neuronal synchrony”. Several corrections have been made helping to clarify some concerns and bringing the results more interesting and reproducible in laboratories using brain slices for ex vivo electrophysiology worldwide. However, there are some key concerns missing that should be properly addressed:

1. In Fig. 6. (or as supplementary), the authors should also show whether the microglia that remain a bit far from the surface shows P2Y₁₂R downregulation and/or gradual depolarization of the resting membrane potential. These experiments will strengthen the results of the differential microglia behaviour across time of incubation and will further extend the knowledge of these effects at more profound circuits in the slice (not only at the slice surface towards the major migration that takes place). In addition, did the authors find differences between the first ultrathin slices and the following? (eg., a gradient of P2Y₁₂R expression between the first till the last one of each time point in the cell body, thin and thick processes).

2. Caption of fig. 8 must be rewritten including in vivo results and mentioning SWRs. As it is written is very generally.

3. In the revised version, the authors missed an important question regarding the different ages used in their study. This is a major concern since the authors may know that the major bulk of ex vivo experiments are performed in brain slices around postnatal days 12-21. The authors have performed some controls at older ages (around P 95) while the approximate range of their experiments is around P 30 – P 80. It is proposed that microglia seem to be less relevant for moderate tissue repair at the slice-cut surfaces as well as for synaptic remodelling and neuronal network formation, at least during the second and third postnatal weeks of hippocampal maturation (See Kann et al., 2016, DOI: 10.1523/JNEUROSCI.0115-22.2022 and Lewen et al., 2020, DOI: 10.1002/jnr.24689). However, some controls performed during these postnatal periods (P8-P14, P15-P21) or at P22-P28 in both areas will strengthen the manuscript novelty, be helpful, and will increase the interest of researchers working on neurodevelopment. There are recent and key works performed in acute brain slices ex vivo from the developing hippocampus and cortex showing

interesting roles for ATP/adenosine in determining plasticity windows that should be revised (see Corlew et al., 2007, DOI: 10.1523/JNEUROSCI.5494-06.2007; Larsen et al., 2014, DOI: 10.1016/j.neuron.2014.07.039, Pérez-Rodríguez et al., 2019, DOI: 10.1093/cercor/bhy194, Falcón-Moya et al., 2020, 10.1038/s41467-020-18024-4, Martínez-Gallego et al., 2022, DOI: 10.1523/JNEUROSCI.0115-22.2022). These studies, and studies from Araque's lab should also be considered in the discussion regarding the possibility that the effects observed in the present work could be driven by astrocytic ATP/adenosine release. In turn, elegant controls blocking the ATP/ADP release from the astrocytes should be shown. Saying this, why do the authors use so different developmental periods even for the same determinations or what is supposed to be a correlated determination? See Andrade-Talavera et al., 2023 doi: 10.1016/j.tins.2023.04.005, where the role of adenosine and ATP during development is extensively revised. In lines 813-822 the authors limit their discussion to microglia in a general way. Whether the experiments suggested appears arduous (which the reviewer considers they are not arduous), at least, the authors should build a short discussion upon the suggested works in this point. The authors may consider that brain slices are widely used for ex vivo study of brain development.

Reviewer #2 (Remarks to the Author):

The authors made an okay argument for each rebuttal to answer my previous concerns. My major issue with the paper is still that it lacked a lot of novelty. I appreciate the great efforts, but showing the results in brain slices with cutting side effect is just not exciting. In that sense, although there is nothing wrong with the science, I won't think the study meets the standard of Nature Communications.

Reviewer #3 (Remarks to the Author):

The authors have made substantial changes to the manuscript including new experiments and improved statistical analyses that have greatly improved the manuscript. I understand that these new experiments are no mean feat and are quite labor intensive. I have a couple more concerns that need addressing in the new additions:

Major comments

1. I very much appreciate the additions to the ATP imaging sections of the manuscript. The added rigor and depth in these experiments makes a great difference to the interpretations and novelty of the findings. The ATP calibrations are particularly helpful. Could the authors add an intermediate and/or lower concentration to their calibration that would help us assess the linearity of the sensor? This is important for assessing the authors interpretation of a bimodal distribution and categorization of flashes vs. surges. Also were all these measurements performed in the same animal/slice? If so they should be repeated in separate animals/experimental days to provide a sense of the variability of the sensor and also expression pattern since the sensor was virally

delivered. It is also a little disingenuous to provide the calibrations in Fig. 4 given that the calibrations were never performed in the transgenic animals (from what I understand from the text) and the sensor expression/localization may be very different. Please either perform the calibrations in the transgenic animals or move this information to Fig. 5. Please also provide some information on how the secondary injury was performed and how the areas for analysis after calibration puffs was chosen.

2. The analysis (and movies suggest) that there may be differential mechanisms at play in responding to the small flashed vs. the large surges. Fig. 5e is hard to interpret. Could the authors standardize the way they are showing the responses such that the onset of the ATP event always happens at a specific time (say 100ms) and the axis end at the same time (say 500ms). The tdTomato changes should also be aligned with a right axis and be the same between all the three conditions shown as it is hard to interpret them with the large change in the x axis for ATP $\Delta F/F$. Also the authors should show at least a representative surge control in the representative images or show the responses to surges and flashes for all three conditions (preferably). Right now it is very hard to compare given the very different event shown for testing of CX3CR1 roles. Also can the authors separate Fig. 5f into flashes and surges?

3. While I do understand the difficulties with repeating experiments in Fig. 1-3 in more animals at many time points I still feel that using multiple slices from the same animal does not constitute independent measurements even though this has been common in past literature. As a compromise the authors should provide animal averages in graphs in a supplementary figure for Fig. 1-3? These can be provided qualitatively without statistical analysis so that the reader can draw their own conclusions. For the cross-lab comparisons in particular, could the authors provide representative images of their systematically randomized measurements they analyzed and more information on how and where they were chosen?

4. The authors show compelling evidence that CX3CR1 is an important mediator of some of their effects. I would have loved to see the authors repeat their experiments in CX3CR1 sufficient animals as most of the experiments presented have CX3CR1 haploinsufficiency. I recognize that this would add an undue burden but the effects of this haploinsufficiency need to be discussed possibly in the section in the discussion about the role of CX3CR1 or in the limitations.

Minor comments

- The improved inclusion of animal sex in the figure legends adds clarity. Please add in line 814 a small statement to the effect of “all male mice were used in this study, with the exception of the data presented in 2c.”
- In the animal section of the methods please add all mice that were used (including the two strains used for ATP measurements) including citations to the papers in which they were first described. It is unclear which tdTomato line the authors used.
- In figures 2h, 2i, 6e, 7d, are the Authors comparing each time point back to control or to the timepoint before it? This is clearly stated in figure 1 but may help with interpretation.

- In figure 8f, if the comparisons are being made to same controls (which it seems like they are) then each panel should be a one-way ANOVA rather than individual tests.
- Line 121-122 Could the authors provide a citation?

REVIEWER COMMENTS

Reviewer #1 (Remarks to the Author): The authors present an improved version of the manuscript formerly entitled “Microglia undergo rapid phenotypic transformation in acute brain slices but remain essential for neuronal synchrony”. Several corrections have been made helping to clarify some concerns and bringing the results more interesting and reproducible in laboratories using brain slices for ex vivo electrophysiology worldwide. However, there are some key concerns missing that should be properly addressed:

We would like to thank the Reviewer for the positive comments on the revised manuscript. We have made efforts to provide an appropriate response to all issues raised. Please find our response in green below and red in the manuscript file.

1. In Fig. 6. (or as supplementary), the authors should also show whether the microglia that remain a bit far from the surface shows P2Y₁₂R downregulation and/or gradual depolarization of the resting membrane potential. These experiments will strengthen the results of the differential microglia behaviour across time of incubation and will further extend the knowledge of these effects at more profound circuits in the slice (not only at the slice surface towards the major migration that takes place). In addition, did the authors find differences between the first ultrathin slices and the following? (eg., a gradient of P2Y₁₂R expression between the first till the last one of each time point in the cell body, thin and thick processes).

We would like to thank the Reviewer for pointing this out. We were not clear enough in the text and in the figure about how exactly the measurements took place. We have indeed sampled the whole width of the slices, and measured receptor levels and electrophysiological properties of microglia not only at the surfaces, but throughout the whole depth of the slices. To clarify this, we have updated the methodological panel in Fig. 6c.

We have also extended the description of the methods section: “After fixation, slices were dehydrated and embedded into resin blocks (1), ultrathin slices were cut onto glass slides (2) and resin etching was followed by post-embedding P2Y₁₂R immunofluorescent labelling (3). Finally, z-stack images were gathered from preparations via high-resolution confocal laser scanning microscopy (4). As each ultrathin section represents the whole cross-section of one acute slice (4), measurement can be done throughout the whole depth range of acute slices within a single image plane.”

We also searched for differences and did not find any gradient along the depth regarding P2Y₁₂R downregulation. This shows that loss of P2Y₁₂R is a general feature of microglial cell bodies and processes across the whole width of acute slices. This information has also been included in the main text (page 20). A representative confocal image (post-embedding approach as

used for quantitative analysis in the manuscript, Holderith et al., 2020) showing microglial P2Y12R levels in two different depths has been added to Supplementary Fig 6.

To assess gradual depolarization, microglial cells were targeted ~50 μm below the slice surface for patch-clamp recordings, as stated in the methods. We deliberately avoided the top ~50 μm of slices (besides the experiments characterizing microglial migration) exactly in order to avoid the top region of slices where the vicinity of the cut surface is expected to cause the most extreme changes in microglial functions. To explain this in the manuscript, we have also modified the schematics in Figure 2 panel b (patch-clamp experiment) and e (microglial morphology quantification), to indicate the depth in which the measurements took place in acute slices.

2. Caption of fig. 8 must be rewritten including in vivo results and mentioning SWRs. As it is written is very generally.

We would like to thank the Reviewer for pointing out this inconsistency. We have amended the figure caption as requested by the Reviewer.

3. In the revised version, the authors missed an important question regarding the different ages used in their study. This is a major concern since the authors may know that the major bulk of ex vivo experiments are performed in brain slices around postnatal days 12-21. The authors have performed some controls at older ages (around P 95) while the approximate range of their experiments is around P 30 – P 80. It is proposed that microglia seem to be less relevant for moderate tissue repair at the slice-cut surfaces as well as for synaptic remodelling and neuronal network formation, at least during the second and third postnatal weeks of hippocampal maturation (See Kann et al., 2016, DOI: 10.1523/JNEUROSCI.0115-22.2022 and Lewen et al., 2020, DOI: 10.1002/jnr.24689). However, some controls performed during these postnatal periods (P8-P14, P15-P21) or at P22-P28 in both areas will strengthen the manuscript novelty, be helpful, and will increase the interest of researchers working on neurodevelopment. There are recent and key works performed in acute brain slices ex vivo from the developing hippocampus and cortex showing interesting roles for ATP/adenosine in determining plasticity windows that should be revised (see Corlew et al., 2007, DOI: 10.1523/JNEUROSCI.5494-06.2007; Larsen et al., 2014, DOI: 10.1016/j.neuron.2014.07.039, Pérez-Rodríguez et al., 2019, DOI: 10.1093/cercor/bhy194, Falcón-Moya et al., 2020, DOI: 10.1038/s41467-020-18024-4, Martínez-Gallego et al., 2022, DOI:

10.1523/JNEUROSCI.0115-22.2022). These studies, and studies from Araque's lab should also be considered in the discussion regarding the possibility that the effects observed in the present work could be driven by astrocytic ATP/adenosine release. In turn, elegant controls blocking the ATP/ADP release from the astrocytes should be shown. Saying this, why do the authors use so different developmental periods even for the same determinations or what is supposed to be a correlated determination? See Andrade-Talavera et al., 2023 doi: 10.1016/j.tins.2023.04.005, where the role of adenosine and ATP during development is extensively revised. In lines 813-822 the authors limit their discussion to microglia in a general way. Whether the experiments suggested appears arduous (which the reviewer considers they are not arduous), at least, the authors should build a short discussion upon the suggested works in this point. The authors may consider that brain slices are widely used for ex vivo study of brain development.

We agree with the Reviewer that microglia have critically important roles during development that can differ substantially from those of adult microglia. In accordance with the complexity of brain development (Thion et al., 2018), we have also described that early developmental formation of somatic purinergic junctions represents an important interface for microglia to monitor the status of immature neurons and control neurodevelopment (Cserep et al., 2022). We also agree that the acute slice model is widely used in developmental brain research. As we highlighted in our previous answer, repeating the whole array of experiments on several developmental age-groups would be clearly beyond the scope of the current study. However, we repeated some key experiments on slices from postnatal day 18-20 mice including measurements on microglial cell body and process translocation, on time-dependent changes of microglial morphology, and ATP activity. Interestingly, we have found that microglia during late postnatal development behave in a very similar manner as in adulthood, further emphasizing the broad impact of the time-dependent changes we observed in this experimental model. We have included the results of cell-body and process translocation and morphology-measurements on Supplementary Figure 1 panel "b" and "d". We thank the Reviewer for this suggestion as we feel that this data further strengthened the study.

In line with this, we performed a new set of two-photon and low magnification epifluorescent ATP imaging studies on brain slices from younger (P19-20) animals. We found similar flashing ATP activity and quickly fading ATP response to secondary mechanical injury, as in slices from adults (please see Suppl. Fig 4a, and also Video 6). These data further confirm that extracellular ATP events, including the newly described sustained flashing ATP activity, represent a general response to tissue injury, occurring similarly in slices from a wide age range (P19-P119). Nevertheless, we fully agree with the reviewer that the flashing ATP events are astroglia related. Beyond literature-based implications, we have seen that these ATP events depict cells with typical astrocytic morphology (and territory; unpublished observations). Also, as discussed during the previous round of revision, there is no reliable way to selectively block astrocyte specific ATP release. The Jing lab (co-authors in the present work) recently addressed this question in depth in a research paper with a completely different focus, performing extensive pharmacological interventions, and suggest a major role for pannexin 1 channels in astroglial ATP release (Chen et al, 2022). On the other hand, others argue for vesicular and volume-regulated anion channel related ATP release (Hatasita et al., 2023). To resolve this complex issue experimentally in the brain slice model is clearly beyond our possibilities. This has been discussed in the manuscript (Page 31).

At the same time, we have extended the discussion, and elaborated on these topics (page 32): “Microglia have critically important roles during development that can differ substantially from those of adult microglia. We have also described that early developmental formation of somatic purinergic junctions represents an important interface for microglia to monitor the status of immature neurons and control neurodevelopment (Cserep et al., 2022). Because the acute slice model is widely used in developmental brain research, we also performed a set of experiments on slices from postnatal mice. Interestingly, we found that microglia during late postnatal development behave in a very similar manner as in adulthood. The immediate ATP release caused by slice cutting or a later direct, secondary injury; the subsequent, decay of the ATP levels at the injury site and the upcoming flashing ATP events were all observed on a similar time- and intensity-scale in slices from a wide (P19-P119) age range. These data underscore the broad impact of the time-dependent changes we observed in this experimental model.”

Furthermore, we extended the discussion (already containing suitable references from the Araque lab) on page 33: “Also, it is important to note, that glial ATP/adenosine release is required for closing/re-opening windows of plasticity during development, both in the hippocampus (Falcon-Moya 2020) and the somatosensory cortex (Gallego 2022; Talavera 2023). Important questions for future research will be how these local, dynamic changes in extracellular ATP, and consequently, adenosine levels could influence microglial and neuronal physiology in *ex vivo* slice preparations and whether such events are sufficient to re-initiate plasticity in the neuronal network in different forms of neuropathologies, *in vivo*.”

Reviewer #2 (Remarks to the Author): The authors made an okay argument for each rebuttal to answer my previous concerns. My major issue with the paper is still that it lacked a lot of novelty. I appreciate the great efforts, but showing the results in brain slices with cutting side effect is just not exciting. In that sense, although there is nothing wrong with the science, I won't think the study meets the standard of Nature Communications.

We would like to thank the careful evaluation of our work and the appreciation of our new experiments / results by the Reviewer. In the revised version of the paper we have made efforts to clarify that the findings presented in this paper go far beyond the objective to study time-dependent changes in acute slices and description of a “cutting side effect”. While we believe that the novel findings presented in the paper would deserve the highest visibility merely from the modelling side (as virtually all studies using the slice model will inherently incorporate injury- and microglia-related affects to the interpretation of the results), we have also identified fundamentally new mechanisms of purinergic microglia-neuron interactions and network physiology by using slice preparation as an experimental model to cause time-dependent changes in microglial states. From the cutting side effect aspect, our novel results propose a major paradigm-shift suggesting to refer to *ex vivo* slice methodology as an injury model, and interpret all results in the literature accordingly. It should be noted that despite the analysis performed across different time points does not look too exciting at first look, such thorough characterization of time-dependent changes after slice preparation concerning synaptic- network-level and microglial properties has not been previously performed. Importantly, we also identify microglia as key modulatory cells affecting complex neuronal network properties across (and despite of) constant changes of their functional states, which is likely to drive further studies and input for broad models of neuronal injury both *ex vivo* and *in vivo*. Furthermore, while ATP has long been known to be a major chemotactic factor and a modulator for microglia, the fact that spontaneous ATP events have such a marked effect on microglial states via P2Y12R and CX3CR1, and in turn that these mediate microglial effects on ripple activity (both *ex vivo* and *in vivo*) have not been addressed in previous studies. We have also provided a point-by-point argument below concerning the novel data sets presented in the manuscript. We are confident that such level of novelty would deserve the expected high visibility and is likely to recruit substantial level of interest from the broad readership of Nature Communications. Novel findings include:

- Cutting-induced and time-dependent changes of microglial states and microglia-neuron interactions, as well as the microglia-dependence of synaptic sprouting (the Reviewer also pointed out this during the first round of comment as an interesting finding)
- The discovery of time-dependent microglial membrane potential changes, and their correlation with phenotype changes in the tissue
- The description of the mechanistic roles of purinergic and fractalkine signaling in the time-dependent microglial cell body / process translocation and morphological changes
- Spontaneous intrinsic ATP-events, and the P2Y12R- and CX3CR1-dependence of microglial recruitment to these spontaneous events has never been observed and described in previous studies
- Microglia-dependence of *ex vivo* SWR and even *in vivo* SWR is entirely novel - the Reviewer also pointed out the importance of this in the previous evaluation round. Taking these recommendations seriously, we addressed the request and performed highly work-demanding and difficult *in vivo* experiments with extensive data analysis, leading to the confirmation of *in vivo* SWR dependence on microglial actions

Taken together, acute slice model is one of the cornerstone experimental paradigms of neuroscience, which grants us a vast amount of knowledge from nanoscale events to complex

neuronal circuits. The mechanisms revealed in this study elevates the acute slice method to further contribute to our understanding of not just injury related pathologies, but pathologies of the CNS that result in strong phenotypic transformation of microglial cells, which is frequently observed in a wide range of diseases. Because the acute slice is a very widely used model, our results are likely to be highly relevant for a broad scientific community and will recruit substantial level of attention. We and many leading experts in the field that we have consulted believe these observations are novel and represent strong foundations to build future experiments on, as it is also of critical importance to reflect back on previous explorations carried out in acute slices.

Reviewer #3 (Remarks to the Author): The authors have made substantial changes to the manuscript including new experiments and improved statistical analyses that have greatly improved the manuscript. I understand that these new experiments are no mean feat and are quite labor intensive. I have a couple more concerns that need addressing in the new additions:

We are grateful to the Reviewer for appreciating our efforts and for raising important questions that helped us to further improve the quality of our work and refine the results. With some additional experiments we are ready to answer all of the queries raised.

Major comments

1. I very much appreciate the additions to the ATP imaging sections of the manuscript. The added rigor and depth in these experiments makes a great difference to the interpretations and novelty of the findings. The ATP calibrations are particularly helpful. Could the authors add an intermediate and/or lower concentration to their calibration that would help us assess the linearity of the sensor? This is important for assessing the authors interpretation of a bimodal distribution and categorization of flashes vs. surges. Also were all these measurements performed in the same animal/slice? If so they should be repeated in separate animals/experimental days to provide a sense of the variability of the sensor and also expression pattern since the sensor was virally delivered. It is also a little disingenuous to provide the calibrations in Fig. 4 given that the calibrations were never performed in the transgenic animals (from what I understand from the text) and the sensor expression/localization may be very different. Please either perform the calibrations in the transgenic animals or move this information to Fig. 5. Please also provide some information on how the secondary injury was performed and how the areas for analysis after calibration puffs was chosen.

We thank the Reviewer for appreciating our ATP imaging studies. To provide more precise data on ATP sensor calibration we performed new experiments, repeatedly delivering gentle ATP puffs (10nM, 50nM, 100nM, 500nM, 1uM, 5uM; five times/each concentration) into distinct sites of separate slices prepared from an adult (P74) mouse. This time, we used slices from a transgenic animal (ATP1.0^{ΔVglut1}), expressing the ATP sensor in VGlut1 positive neurons, identical to those mice we used in the original experiments to gain data on Fig 4. Importantly, the ATP sensor response to 100nM ATP was very similar to that we measured in slices prepared after viral ATP sensor delivery, while the 1uM ATP revealed somewhat lower response in the transgenic slices. We inserted the new 50-100-1000 nM ATP calibration data

into Fig 4i and added a new supplementary graph (Suppl. Fig 4d), showing that the ATP sensor behaves linearly between the concentrations that we used for calibration. We also changed the text in the manuscript accordingly (page 14) and refreshed the Methods section (page 55). We thank the Reviewer for drawing our attention to this matter. In addition, we would like to mention that in-depth sensor calibration had been published in the paper introducing the novel ATP sensor, showing a dose response curve and a calculated $EC_{50}=77nM$ ATP value in primary neurons (Wu et al, 2022, Neuron, Fig 2I), which nicely fits our results.

We also performed additional experiments on secondary injury, cutting slices with a sterile blade and starting imaging almost immediately (no more than ~1-2 min) after injury. For this purpose, slices incubated for several hours (5+/-1,7h) after slice preparation were used from either P74 or younger, P20 ATP1.0 Δ^{Vglut1} animals. The videos, in agreement with previous data on virally labelled slices, show an intense ATP signal after injury which decays fast compared to non-injured regions or non-injured slices. Please find the renewed Suppl. Fig 4a and Supplementary Video 6. The Methods section in the manuscript has been upgraded accordingly.

2. The analysis (and movies) suggest that there may be differential mechanisms at play in responding to the small flashed vs. the large surges. Fig. 5e is hard to interpret. Could the

authors standardize the way they are showing the responses such that the onset of the ATP event always happens at a specific time (say 100ms) and the axis end at the same time (say 500ms). The tdTomato changes should also be aligned with a right axis and be the same between all the three conditions shown as it is hard to interpret them with the large change in the x axis for ATP $\Delta F/F$. Also the authors should show at least a representative surge control in the representative images or show the responses to surges and flashes for all three conditions (preferably). Right now it is very hard to compare given the very different event shown for testing of CX3CR1 roles. Also can the authors separate Fig. 5f into flashes and surges?

We have resized and re-measured the original representative images and standardized the corresponding graphs on Fig 5e for better transparency and comparability.

Also, we added images and graphs to illustrate the dimmer/faster/smaller events (flashes) in comparison to the brighter/slower/bigger surges shown in the previous version. Also, the Supplementary Video 7 file has been complemented accordingly. We have not implemented another axis for the tdTomato signal on the right side of the plots, as they would be the same as in the left side.

We have also separated the data shown on Fig 5f to represent the change in %MFI separately for flashes and surges as the Reviewer suggested. While PSB intervention seemed to significantly reduce directed movements towards both flashes and surges, AZD intervention indeed shows a difference and had a larger negative effect on movements toward flashes than towards surges. This is also consistent with the discrete quantification of directed movements on panel "h". We decided to not separate the latency of movements on panel "f", as we did not find any significant difference between ctrl and PSB/AZD treated latencies after the separation. These panels are now included in the new version of the manuscript and the text in the results and figure captions have also been modified accordingly: "Importantly, microglial processes were rapidly (within 1-3 min) recruited to spontaneously emerging focal ATP events with a directed movement (dm) towards the center of flashes/surges (Suppl. Video 7, Fig 5d-

e). We quantified these microglial actions as the change in the tdTomato signal ($\Delta\%$ MFI, see: Methods) measured in the area (ROI) of flashes (Suppl.Fig.4e) and surges (Fig.5e) before and after a given ATP release event. Of note, blockade of either P2Y12R (PSB) or CX3CR1 receptors (AZD) largely prevented microglia process recruitment to focal ATP events categorized as flashes (Fig 5f, left), while directed movements towards surges seemed to be less affected by AZD treatment.”

3. While I do understand the difficulties with repeating experiments in Fig. 1-3 in more animals at many time points I still feel that using multiple slices from the same animal does not constitute independent measurements even though this has been common in past literature. As a compromise the authors should provide animal averages in graphs in a supplementary figure for Fig. 1-3? These can be provided qualitatively without statistical analysis so that the reader can draw their own conclusions. For the cross-lab comparisons in particular, could the authors provide representative images of their systematically randomized measurements they analyzed and more information on how and where they were chosen?

We would like to thank the Reviewer for raising this point. We completely agree that animal-to-animal variation can substantially add to overall variation of data, thus the correct way to perform these experiments is to examine slices from individual animals over all experimental conditions, in our case all timepoints. As we have observed the same time-dependent microglial behavior in all animals, we could pool these data. With all the necessary control experiments, and the comparisons between different laboratories, cutting techniques, room temperature cutting, thinner and thicker slices, different age groups (now including late postnatal mice), two brain areas (neocortex and hippocampus), different mouse lines (C57Bl6, CX3CR1-GFP) we are confident that the observed microglial behavior is a very basic and general feature of acute slices. Nevertheless, we prepared a new supplementary figure, where we plotted the most important measurements from figures 1-3 while marking the different animals. We have added the following text to the Results: “As microglia have critically important roles during development that can differ substantially from those of adult microglia, and the acute slice model is also widely used in developmental studies, we have repeated some experiments on slices from postnatal day 18-20 (P18-20) mice. We found that time-dependent microglial cell body and process translocation (Suppl. Fig 1b), and also time-dependent changes of microglial morphology (Suppl. Fig 1d) happen in slices from mice in late postnatal developmental phase in a very similar manner as in adulthood, underscoring

the broad impact of the cutting-induced changes we observed. (See also averages of individual animals on Suppl. Fig 2).”

Furthermore, we have added representative images to Figure 3 for the cross-lab comparisons as Fig 3 d.

Systematic random sampling was achieved by placing equal sized ROIs on the same positions of the sections, based only on the tissue outline (i.e. without seeing the microglial labelling). In the morphology measurements the whole image stacks were measured (the total depth of ~100 um from top to bottom of the re-sectioned slices), and the localization of areas were guided by DAPI signal to be placed in the same region both for hippocampus and cortex, as described in the schematics.

4. The authors show compelling evidence that CX3CR1 is an important mediator of some of their effects. I would have loved to see the authors repeat their experiments in CX3CR1 sufficient animals as most of the experiments presented have CX3CR1 haploinsufficiency. I recognize that this would add an undue burden but the effects of this haploinsufficiency need to be discussed possibly in the section in the discussion about the role of CX3CR1 or in the limitations.

We would like to thank the Reviewer for pointing out this important topic. We are aware of the fact that the literature generally uses CX3CR1-haploinsufficient mice as controls, however these mice generally show mild phenotypic changes. To circumvent this problem, we have performed the key experiments on CX3CR1 sufficient mice, or, in selected experiments, we also used acute pharmacological inhibition of CX3CR1. In the experiments aimed to compare different laboratories, all mice were wild-type C57Bl6, and these slices showed the same microglial cell-body and process translocation, and also the time-dependent morphological changes as those from CX3CR1-GFP mice (Fig. 3). The experiments with ATP-sensors were either performed using CX3CR1-sufficient mice (Fig. 4), or - when using AAV-injection into CX3CR1-cre/tdTomato mice - we applied acute pharmacological inhibition of CX3CR1 with AZD8797 (Fig. 5). In the *ex vivo* / *in vivo* SWR measurements we also used wild-type mice as controls (Fig. 8), meaning that all key results were confirmed in CX3CR1 sufficient mice. Nevertheless, for clarity, we have included the following sentence in the discussion section: “Depending on the experimental model used, CX3CR1 ^{+/-} mice in previous studies showed largely similar phenotypes to WT mice or intermediate phenotypes between WT and KO (Gyoneva et al., 2019). To consider this issue in our study, we have used either CX3CR1 ^{+/-} mice as a microglia-reporter line – in these cases the experiments were also repeated on CX3CR1 sufficient mice - or we used acute pharmacological inhibition of CX3CR1; while in experiments aimed to investigate the involvement of fractalkine signaling we compared CX3CR1-KO to wild-type mice.”

Because the extensive investigations revealed several unexpected mechanisms related to purinergic signaling, we decided to change the title to better reflect the involvement of these mechanisms. The new title is: "Microglia are essential for neuronal synchrony despite endogenous ATP-related phenotypic transformation in acute brain slices".

Minor comments

- The improved inclusion of animal sex in the figure legends adds clarity. Please add in line 814 a small statement to the effect of "all male mice were used in this study, with the exception of the data presented in 2c."

We have added the requested text to the manuscript.

- In the animal section of the methods please add all mice that were used (including the two strains used for ATP measurements) including citations to the papers in which they were first described. It is unclear which tdTomato line the authors used.

We would like to thank the Reviewer for bringing our attention to this point, we have extended the methods description accordingly.

- In figures 2h, 2i, 6e, 7d, are the Authors comparing each time point back to control or to the timepoint before it? This is clearly stated in figure 1 but may help with interpretation.

We are grateful for pointing out this unclarity, each timepoint is compared to the 0 min values. We have amended the figure legends accordingly.

- In figure 8f, if the comparisons are being made to same controls (which it seems like they are) then each panel should be a one-way ANOVA rather than individual tests.

We apologize if this was not entirely clear - each group had its own control group. The control slices for each comparison are prepared from littermates and are from the same age range, and control/experimental slices are recorded simultaneously in the same chamber. This is visible from the similar, but not identical variance of control data points, and also stated in the methods section. We found it necessary to have a dedicated control group for each comparison and measure ctrl vs. depleted/KO slices simultaneously, to exclude artefacts that might arise due to differences in experimental conditions. Nevertheless, we have added two lines in the figure legends and methods to clarify the issue.

- Line 121-122 Could the authors provide a citation?

Thank you for marking this point, we have added the required references.

REVIEWERS' COMMENTS

Reviewer #1 (Remarks to the Author):

Authors have addressed the remaining concerns

Reviewer #3 (Remarks to the Author):

The authors have satisfied all my concerns.